# Postictal behavioural impairments are due to a severe prolonged hypoperfusion/ hypoxia event that is COX-2 dependent

Jordan S Farrell[1], Ismael Gaxiola-Valdez[1], Marshal D Wolff[1], Laurence S David[1], Haruna I Dika[1,2], Bryce L Geeraert[1], X Rachel Wang[1], Shaily Singh[1], Simon C Spanswick[1], Jeff F Dunn[1], Michael C Antle[1], Paolo Federico[1*†], G Campbell Teskey[1*†]

[1]Hotchkiss Brain Institute, Cumming School of Medicine, University of Calgary, Calgary, Canada; [2]Department of Physiology, Catholic University of Health and Allied Sciences, Mwanza, Tanzania

**Abstract** Seizures are often followed by sensory, cognitive or motor impairments during the postictal phase that show striking similarity to transient hypoxic/ischemic attacks. Here we show that seizures result in a severe hypoxic attack confined to the postictal period. We measured brain oxygenation in localized areas from freely-moving rodents and discovered a severe hypoxic event ($pO_2 < 10$ mmHg) after the termination of seizures. This event lasted over an hour, is mediated by hypoperfusion, generalizes to people with epilepsy, and is attenuated by inhibiting cyclooxygenase-2 or L-type calcium channels. Using inhibitors of these targets we separated the seizure from the resulting severe hypoxia and show that structure specific postictal memory and behavioral impairments are the consequence of this severe hypoperfusion/hypoxic event. Thus, epilepsy is much more than a disease hallmarked by seizures, since the occurrence of postictal hypoperfusion/hypoxia results in a separate set of neurological consequences that are currently not being treated and are preventable.

*For correspondence: pfederic@ucalgary.ca (PF); gteskey@ucalgary.ca (GCT)

†These authors contributed equally to this work

**Competing interests:** The authors declare that no competing interests exist.

## Introduction

For proper neuronal functioning, brain tissue oxygen levels are normally maintained through exquisite regulation of blood supply (*Erecińska and Silver, 2001*). However, during neurological events when brain tissue oxygen levels fall below the severe hypoxic threshold ($pO_2 < 10$ mmHg), neuronal dysfunction and behavioural disturbances are observed (*Farrar, 1991*; *van den Brink et al., 2000*; *Maloney-Wilensky et al., 2009*). Following termination of a seizure, behavioral impairments related to the specific brain structures that participated in the seizure are expressed (*Leung et al., 2000*; *Gallmetzer et al., 2004*). Todd's Paresis, for example, specifically refers to moderate to severe motor weakness following seizures and typically subsides within a few hours (*Todd, 1849*). The symptoms are so similar to ischemic stroke that Todd's paresis is often misdiagnosed (*Mathews et al., 2008*; *Masterson et al., 2009*). The occurrence of these behavioral impairments have been attributed to the affected cortex being 'exhausted' (*Franck and Pitres, 1878*) or silenced due to increased inhibition (*Gowers, 1901*), but these conjectures are not supported. Despite being first described over 160 years ago, the cause of postictal behavioral disturbances has not been determined. A few case studies of persons experiencing Todd's paresis demonstrated that the affected cortex was inadequately perfused during this period (*Mathews et al., 2008*; *Yarnell and Paralysis, 1975*; *Rupprecht et al., 2010*). Thus, we hypothesized that following termination of a seizure, a long-lasting and severe hypoxic episode would occur and manifest as behavioral dysfunction.

**eLife digest** It has long been known that after an epileptic seizure, individuals often experience an extended period of impairments that affect how the brain works. Because the brain is organized so that specific tasks happen in particular areas, seizures that affect areas of the brain that control movement are often followed by muscle weakness. Likewise, amnesia may follow a seizure that affects brain areas involved in memory. While these events reduce quality of life in people with epilepsy, they have gone untreated because we did not understand what occurs in the brain after seizures.

It has been observed that the impairments that follow seizures are similar to those that follow strokes, where for a period of time blood flow to certain areas of the brain is restricted and these areas are starved of oxygen. Following a seizure, is there a local stroke-like event that is responsible for the behavioural and memory impairments?

To address this question, Farrell et al. studied blood flow in the brains of mice, rats and human volunteers with epilepsy. The experiments show that after an epileptic seizure, blood vessels become narrower, which reduces blood supply to the areas of the brain involved in the seizure and dramatically reduces oxygen levels in those same areas. Using drugs to block the activity of an enzyme called cyclooxygenase-2 or other proteins called L-type calcium channels prevented both the oxygen shortage and the behavioural impairments that follow seizures. Thus, people with epilepsy are experiencing stroke-like events after seizures that they should be able to avoid with simple medical treatments.

In the future, clinical research should determine how effective these treatments are in people with epilepsy. Other experiments should reveal if a shortage of oxygen after a seizure causes noticeable brain damage and the long-term behavioural problems that are often associated with epilepsy.

This simple idea is intriguing, testable and provides a critical new insight into this broad reaching neurological disease by explaining some of the detrimental consequences associated with epilepsy. But to date, a continuous and detailed examination of local tissue oxygenation and blood flow following seizures has not been carried out.

Most investigations of the hemodynamics associated with seizures have concentrated on changes before or during the seizure and not during the postictal period. It is well established that there is a dramatic, transient increase in blood flow during seizures (*Gibbs et al., 1934*; *Penfield et al., 1939*) with a corresponding decrease in oxygenation that quickly recovers (*Bahar et al., 2006*; *Suh et al., 2006*). The few studies that investigated changes in blood flow during the postictal period have not yielded consistent observations with reports of local hypoperfusion (*Rowe et al., 1991*; *Newton et al., 1992*; *Leonhardt et al., 2005*) and hyperperfusion (*Fong et al., 2000*; *Tatlidil, 2000*; *Hassan et al., 2012*). These inconsistencies were likely due to the variable time-points after the seizure when blood flow was measured. To circumvent this problem we systematically investigated local oxygen levels and blood flow following evoked and spontaneous seizures in rats and mice. We then relied on those results to guide our clinical study and used arterial spin labeling (ASL) MRI to measure postictal hypoperfusion following spontaneous seizures in people with epilepsy.

We further hypothesized that the enzyme cyclooxygenase-2 (COX-2), which is primarily responsible for catalyzing arachidonic acid to the PGH2-derived prostanoids (*Hla and Neilson, 1992*) and plays a major role in neurovascular coupling (*Niwa et al., 2000*; *Lecrux et al., 2011*; *Lacroix et al., 2015*), would mediate postictal hypoperfusion/hypoxia. We reasoned that electrographic seizures would engage this system and lead to a pronounced vasoconstriction and subsequent severe hypoxia. Downstream of this, we anticipated that elevated free calcium in vascular smooth muscle, particularly calcium conducted by L-type channels (*Putney and McKay, 1999*), would sustain this vasoconstriction. Here we demonstrate that blocking these pathways prevents postictal severe hypoperfusion/hypoxia and spotlights its contribution to postictal behavioral impairments.

## Results

### Long-lasting severe hypoxia follows spontaneous and evoked seizures

A bipolar electrode and oxygen-sensing probe (optode) were chronically implanted into CA1 and CA3, respectively, of the dorsal hippocampus (*Figure 1L*). The optode continuously recorded the partial pressure of oxygen (pO$_2$ in mmHg) in awake, freely-moving rats before, during and, most importantly, after brief electrographic seizures over several weeks. During typical behavioral states, hippocampal pO$_2$ levels vary within the normoxia range between 18 and 30 mmHg (*Figure 1A*), as previously reported (*Erecińska and Silver, 2001*). 10 mmHg oxygen was chosen as the threshold for defining severe hypoxia since several independent studies have demonstrated that pO$_2$ levels at or below 10 mmHg cause significant changes to cellular physiology and brain injury. Hypoxia-dependent gene expression via Hypoxia-Inducible Factor 1 transcriptional activation occurs as an exponential function with respect to decreasing pO$_2$ (*Jiang et al., 1996*). In a HeLa cell culture system, 10–15 mmHg was determined to be the half-maximal response and since the steep portion of the exponential curve is at lower pO$_2$ levels, most gene expression occurs below this threshold. Restricting middle cerebral artery blood flow in the cat led to cell death at a threshold of 7–8 mmHg (*Farrar, 1991*). Furthermore, both the depth and duration of severe hypoxia following traumatic brain injury are good predictors for clinical outcome (*van den Brink et al., 2000*; *Maloney-Wilensky et al., 2009*). 10 min of pO$_2$ below 10 mmHg was associated with increased risk of death, and further increases with lower pO$_2$ and longer duration of severe hypoxia (*van den Brink et al., 2000*). A systematic review of clinical traumatic brain injury found that 10 mmHg was a suitable threshold for defining severe hypoxia associated with worse outcomes (*Maloney-Wilensky et al., 2009*). In defining tumor hypoxia, which is thought to exacerbate cancer pathophysiology, 8–10 mmHg is the suggested threshold (*Höckel and Vaupel, 2001*). Thus, 10 mmHg is a reasonable threshold for defining severe hypoxia and integrating the entire area below 10mmHg combines the depth and duration to reveal the total hypoxic burden.

Oxygen levels were then monitored in the intrahippocampal kainate model of temporal lobe epilepsy (*Rattka et al., 2013*) before, during, and after a spontaneous seizure. We measured a brief and small drop in hippocampal oxygenation at seizure onset and a subsequent increase in oxygenation during the electrographic seizure. However, oxygen levels dropped precipitously to below the severe hypoxic level (pO$_2$ < 10 mmHg) following seizure termination (*Figure 1B*). Even more striking was that the severe hypoxia was sustained for an hour. We then systematically examined this phenomenon in a higher throughput model of evoked focal seizures: electrical kindling.

Using electrical kindling we elicited an afterdischarge (i.e. electrographic seizure) by applying brief stimulation (1 ms pulse widths at 60 Hz for 1 s) through the chronically implanted bipolar electrode. A brief dip in oxygenation at seizure onset and subsequent increase was observed (*Figure 1C*). Importantly, the severe hypoxic event following a kindled seizure was markedly similar in magnitude and duration compared to spontaneous seizures (*Figure 1C*). Since no substantive differences in postictal hypoxia were observed between the two models, we chose to evoke afterdischarges with kindling stimulation as our principal model for subsequent experiments to maximize data collection efforts.

Daily repeated stimulation of the dorsal hippocampus gave rise to a wide range of primary afterdischarge durations across rats (10–50 s) and allowed us to determine that there is a strong positive and linear relationship between the duration of the hippocampal seizures and the severity of postictal hypoxia, as revealed by integrating the area below 10 mmHg (*Figure 1D*).

To provide convergent evidence for this postictal hypoxia phenomenon, we also used an immunohistological staining technique based on pimonidazole (Hypoxyprobe™-1), which selectively labels hypoxic cells (*Chapman et al., 1981*; *Höckel and Vaupel, 2001*). Rats were injected with pimonidazole and afterdischarges elicited. Since postictal severe hypoxia in the rat hippocampus lasted a mean of 73.1 ± 7.8 min (*Figure 1N*), we examined brains one hour post-seizure. We observed significantly increased numbers of pimonidazole-labeled cells in the dentate hilus and CA3 regions (*Figure 1E–K*) relative to non-seizure controls. Thus, two different and unrelated techniques identified a severe hypoxic period following brief seizures.

We then asked the question whether the phenomenon could be observed in a different structure. Rats were chronically implanted with an electrode in the corpus callosum to elicit neocortical afterdischarges and an optode was implanted into layer V of motor neocortex (*Figure 1M*). Postictal severe

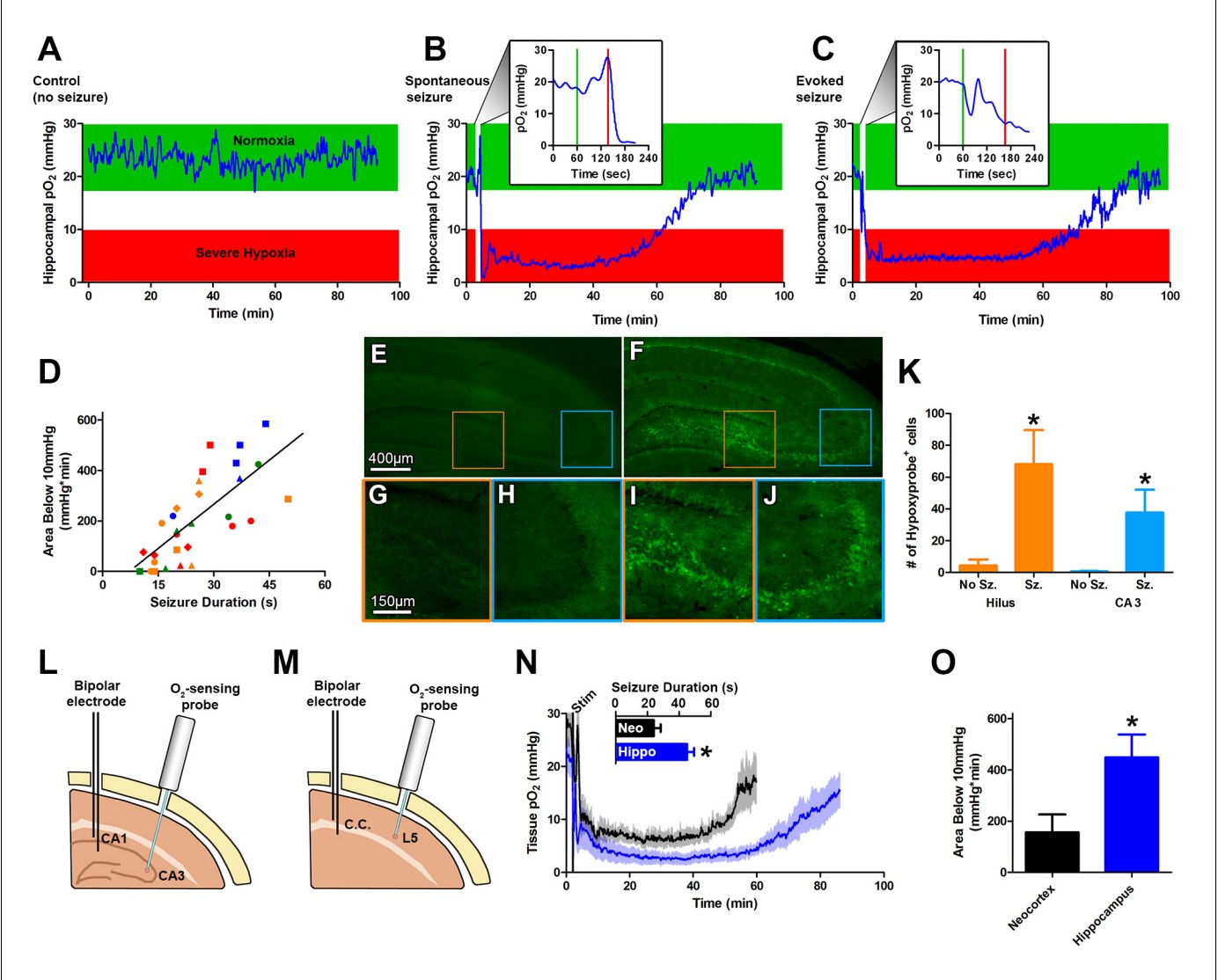

**Figure 1.** Seizures induce severe postictal hypoxia. (A) Local tissue oxygenation in the hippocampus of an awake, freely-moving rat (blue). Green denotes normoxia while red denotes severe hypoxia. (B) Representative oxygen profile before, during, and after a spontaneous seizure. The inset expands the time-scale during the seizure with the green and red lines denoting the beginning and end of an 80 s seizure. This inset corresponds to the white vertical block near the beginning of the full oxygen recording. (C) Representative oxygen profile before, during, and after a 106 s electrically kindled seizure. (D) Scatterplot of the relationship between the duration of kindled seizures in the dorsal hippocampus (primary afterdischarge) and the degree of severe hypoxia expressed as the total area below the severe hypoxic threshold (10.0 mmHg) by time (min). Symbols with both the same colour and shape are from the same animal (n = 14). The line of best fit (y = 11.85x+7.57) is indicated. R square = 0.55, p<0.0001. (E–J) Hypoxyprobe immunohistochemistry. (E) Representative image from control rat with close-ups of CA3 (H) and hilus (G). (F) Representative image following a seizure with close-ups of CA3 (J) and hilus (I). Scale bar for (E,F) = 400 µm. Scale bar for (G–I) = 150 µm. (K) Densely stained neurons in the hilus and CA3 were quantified. There are significantly more stained cells in the hilus and CA3 following seizures. Data are mean ± SEM. *p<0.05 (t-test). (L–N) Comparing hippocampus and neocortex. (L) Location of chronic hippocampal implants. (M) Location of chronic neocortical implants. Bipolar electrodes were used for stimulating and recording seizure and O2 sensors for continuous oxygen recordings. C.C. is corpus callosum, L5 is layer 5. (N) Inset displays mean seizure duration ±SEM (n = 5). Hippocampus had significantly longer seizures. *p<0.05. The mean pO2 (opaque) ±SEM (transparent) over time recorded in motor neocortex dorsal hippocampus. (O) Quantification of (N). Hypoxia was more severe in the hippocampus relative to neocortex as assessed by the area below 10mmHg. *p<0.05 (t-test).

The following figure supplement is available for figure 1:

**Figure supplement 1.** Postictal severe hypoxia generalizes to other seizure models.

hypoxia was also present in the motor cortex where the typically shorter neocortical afterdischarges gave rise to higher minimum $pO_2$ and a more rapid return to baseline oxygen levels, relative to hippocampus (*Figure 1N*). Quantification of severe hypoxia in these two structures revealed significantly worse severe hypoxia in the hippocampus (*Figure 1O*), likely due to longer seizure durations.

To further ensure that postictal hypoxia was not specific to the method of seizure induction we also tested two other experimentally-induced seizure models. Maximal electroconvulsive shock (MES) which models generalized convulsions (*Young et al., 2006*) and 3 Hz electrical stimulation, which drives epileptiform activity and behavioral seizures (*Teskey and Racine, 1993*) both induced a long-lasting hypoxic event in the hippocampus (*Figure 1—figure supplement 1*) similar to those observed following spontaneous or electrical kindled seizures. We conclude that a postictal severe hypoxic event follows epileptiform seizures regardless of the eliciting technique.

## Hypoperfusion mediates postictal severe hypoxia

Severe hypoxic events are often the consequence of inadequate blood flow mediated by vasoconstriction and determining the contribution of reduced blood flow to hypoxia can identify new routes for intervention. We first determined the relationship between local tissue hypoxia and blood flow using an implantable laser Doppler flowmetry (LDF) probe to measure relative changes in hippocampal blood flow while simultaneously monitoring $pO_2$ levels in the CA3 region of the rat dorsal hippocampus (*Figure 2A*). *Figure 2B* displays concurrent measures of mean tissue $pO_2$ with blood flow. Blood flow increased during afterdischarges, which was consistent with previous observations (*Metzger, 1977*). Importantly, the postictal severe hypoxia is accompanied by reduced blood flow to $53.0 \pm 8.3\%$ relative to baseline between 40 and 60 min post-seizure. Furthermore, blood flow recovered on a similar time-scale to tissue oxygenation indicating that the postictal severe hypoxia is, in part, related to hypoperfusion.

If the postictal hypoxia is due to a vascular event then blocking calcium conductance should reduce vasoconstriction and prevent hypoxia. Pre-treatment with the L-type calcium channel antagonist nifedipine 30 min prior to seizure elicitation did not alter the afterdischarge duration (*Figure 2C*) indicating that nifedipine effects can be attributed to a neurovascular mechanism. We observed higher $pO_2$ levels at baseline (+9.0 mmHg, p<0.01, paired ANOVA), throughout the postictal period, and an earlier return to baseline relative to vehicle (*Figure 2C*). Moreover, nifedipine pre-treatment significantly attenuated the area below the severe hypoxic threshold ($pO_2 < 10$ mmHg) (*Figure 2D*). We also observed that nifedipine was effective at reducing postictal severe hypoxia when administered immediately following termination of the afterdischarge (*Figure 2—figure supplement 1A,B*). This is not surprising since L-type calcium channels play a key role in the tonic phase of vascular smooth muscle contraction (*Putney and McKay, 1999*). These results provide key evidence that hypoxia is largely mediated by hypoperfusion and that L-type calcium channel activity plays a role in sustaining this response.

While laser Doppler flowmetry provides inferential evidence that blood flow was reduced postictally, we sought a direct measure of changes in blood vessel diameter following epileptiform-like activity in the hippocampus. In young (P25-P40) freely moving rats, we first determined that 2 min of 3 Hz stimulation resulted in a remarkably similar profile of severe hypoxia compared to the standard 1 s stimulation evoked afterdischarges (*Figure 2E,F*). We then moved the 3 Hz stimulation protocol to an acute hippocampal slice preparation (*Figure 2G*) and imaged hippocampal arterioles in the synaptic layer of CA1 (stratum radiatum) before, during, and after Schaffer collateral stimulation (*Figure 2H,I*). We observed sustained $13.8 \pm 4.0\%$ arteriole constriction following stimulation (60–90 min post-stim, *Figure 2J*). Poiseuille's law dictates that a 13.8% reduction in diameter would lead to blood flow that is 55.2% of baseline, which correlates well with our laser Doppler flowmetry measurements ($53.0 \pm 8.3\%$ at 40–60 min, *Figure 2B*). Pretreatment with nifedipine (50 µM) caused significant vessel dilation ($108.4 \pm 2.9\%$ relative to baseline, p<0.05, paired t-test) and prevented 3 Hz stimulation-induced vessel constriction (*Figure 2K*). Our data obtained using laser Doppler flowmetry, nifedipine treatment, and arteriole constriction in slice all provided convergent evidence that severe hypoxia following seizures is mediated by vasoconstriction and hypoperfusion.

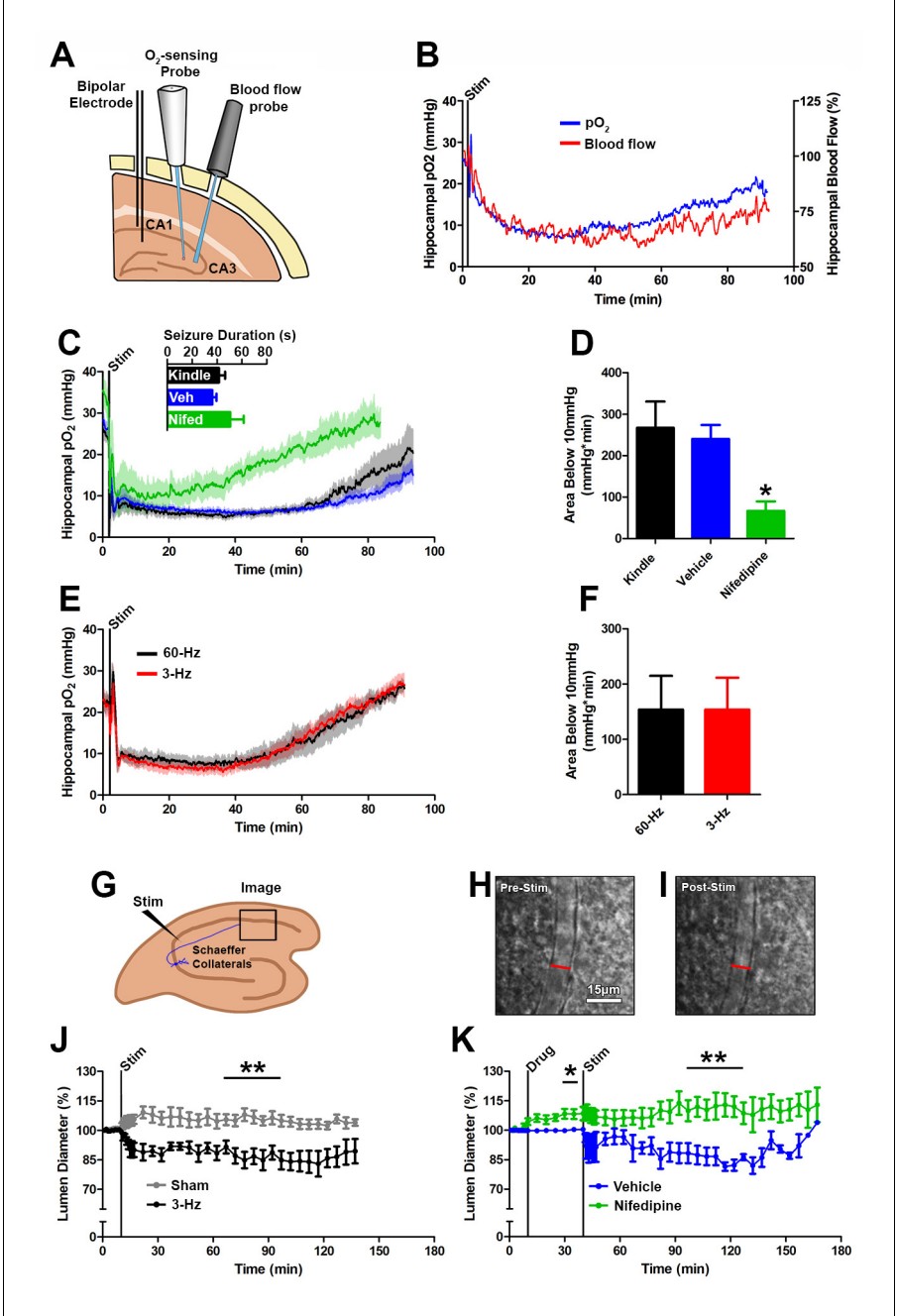

**Figure 2.** Seizures cause postictal vessel constriction in an in vitro preparation and reduced blood flow (hypoperfusion) in vivo. (A) Location of implants for simultaneous blood flow and $pO_2$ recordings. These two probes were placed at opposing angles to leave room for cable attachment. (B) The simultaneous measurement of mean blood flow and mean $pO_2$ in the hippocampus following brief seizures (n = 5). (C) Nifedipine pre-treatment (15 mg/kg) caused an elevation of baseline $pO_2$, inhibited severe hypoxia, and increased the rate of recovery (n = 5). Inset reveals no difference in seizure duration. (D) Quantification of (C). Nifedipine pre-treatment reduced the amount of severe hypoxia (area below 10mmHg). *$p<0.05$ (within-subject ANOVA). (E) Validation of 3 Hz stimulation in young rats (P28–P35). Mean oxygen profiles following standard kindling and 3 Hz stimulation (n = 6). (F) Quantification of (E). No significant differences were found in the amount of severe of hypoxia (area below 10 mmHg). (G) Acute hippocampal in vitro slice preparation for measuring postictal vessel constriction. 3 Hz stimulation was applied to the Schaeffer collaterals and imaging was captured in stratum radiatum of CA1. (H,I) Representative images of CA1 arteriole pre-stimulation (H) and post-stimulation (I). Scale bar for (H) and (I) is 15 μm. (J) Lumen diameter over time in stimulated and sham controls. Mean lumen diameter is reduced following 3

*Figure 2 continued on next page*

*Figure 2 continued*
Hz stimulation (n = 11) relative to sham (n = 8) between 60 and 90 min post-stim. Data displayed as mean ± SEM. **p=0.001 (t-test). (K) 30 min following nifedipine (50 μM; n = 7) or vehicle (n = 5) application slices were stimulated. Mean lumen diameter was significantly different between 60 and 90 min post-stim. **p<0.01.
The following figure supplement is available for figure 2:

**Figure supplement 1.** Post-seizure administration of nifedipine prevents severe postictal hypoxia.

## Postictal hypoperfusion in clinical epilepsy

Based on our time-course data generated from model systems, we asked whether there was postictal hypoperfusion within one hour of a spontaneous ictal event in epilepsy patients. All patients had intractable focal epilepsy and the mean age was 32 years (range 20–42 years). Two patients were excluded from the study due to excessive motion during the acquisition of postictal ASL images, leaving 10 patients for analysis (*Table 1*). The mean age at seizure onset was 12.3 years (range one month-29 years). Four patients had normal MRI scans, three had malformation of cortical development, one had an occipital cavernoma, one had nonspecific changes and one had postsurgical changes as a result of previous surgery. Cerebral blood flow (CBF) was quantified in postictal and baseline (interictal, >24 hr without a seizure) ASL scans. Postictal ASL images were subtracted from baseline scans to reveal areas of hypoperfusion. Maximal changes in blood flow were quantified in mL/100 g/min and the brain area where this change occurred was compared to ictal EEG to determine concordance between local hypoperfusion and seizure activity (*Table 2*).

Maximal postictal cerebral blood flow reductions of at least 10 mL/100 g/min were seen in 8 of 10 subjects (*Table 2*). Looking at the region of interest (ROI) for all 10 subjects, we observed a mean baseline CBF of 59.0 ± 4.4 mL/100 g/ and mean CBF of 42.1 ± 3.4 mL/100 g/min following seizures, for a difference of 16.9 ± 4.1 mL/100 g/min or 26.6 ± 6.0% from baseline (*Figure 3—figure supplement 1A*). Furthermore, the magnitude of hypoperfusion was positively correlated with seizure duration (*Figure 2—figure supplement 1B*), just as we saw with the magnitude of hypoxia and seizure duration in the rat (*Figure 1D*). In the two subjects without observable hypoperfusion of >10 mL/100 g/min, the seizure durations were notably shorter and may account for the absence of hypoperfusion. Excluding these two patients, CBF of 65.0 ± 3.6% relative to baseline was observed in the postictal scan. While we observed a greater change in the rat (53.0 ± 8.3% relative to baseline), the ROI for ASL quantification also included adjacent confluent voxels of less magnitude, which would decrease the reported change, and the relative heterogeneity of subjects should be noted.

This heterogeneity of subjects, however, allowed us to investigate the overlap between the brain areas of maximal hypoperfusion and areas of seizure activity to determine the specificity of this phenomenon. Two examples are shown in *Figure 3* (ASL 001, 010) and provide qualitative confirmation

**Table 1.** Characteristics of patients recruited to ASL study.

| Patient | Age | Age at Seizure Onset | Structural MRI findings |
|---------|-----|----------------------|-------------------------|
| ASL-001 | 38 | 34 | Normal |
| ASL-003 | 41 | 13 | Lesion in the right superior temporal gyrus, resolving? |
| ASL-004 | 25 | 5 | Normal |
| ASL-006 | 33 | 7 | Normal |
| ASL-007 | 33 | 29 | Bilateral subependymal heterotopias along lateral ventricle right > left |
| ASL-008 | 42 | 22 | Right amygdala enlargement, Right occipital cavernoma |
| ASL-009 | 40 | 9 | Normal |
| ASL-010 | 26 | 1 | Left frontal malformation of cortical development |
| ASL-011 | 22 | 14 | Normal |
| ASL-012 | 20 | 0.5 | Postsurgical changes in the right frontal lobe, right anterior temporal lobectomy |

**Table 2.** Concordance of postictal ASL hypoperfusion with brain areas involved in the seizure. EEG localization of ictal activity was determined by an Epileptologist based on scalp EEG. GTC denotes when a generalized tonic-clonic seizure occurred. Areas of maximal hypoperfusion were identified by a blinded reviewer and Regions of Interest were drawn around those areas and adjacent confluent voxels for quantification (*Figure 2—figure supplement 1A,B*). Concordance was noted if ictal activity overlapped with the area of maximal hypoperfusion.

| Patient | EEG localization of seizure [duration (S)] | Area of maximal hypoperfusion for ASL quantification | ROI volume (cm$^3$) | Concordant |
|---|---|---|---|---|
| ASL-001 | Left temporal [71] | Left temporal | 2.52 | yes |
| ASL-003 | Unclear [26] | No Change | 1.45 | n/a |
| ASL-004 | Left frontocentral [87] | Left superior posterior temporal | 0.31 | yes |
| ASL-006 | Left fronotemporal [114] | Left insula, left anterior temporal | 1.46 | yes |
| ASL-007 | GTC, Left hemisphere. Maximal posterior temporo-parietal [64] | Multifocal bihemispheric | 1.35 | yes |
| ASL-008 | GTC, Right temporo-occipital [155] | Multifocal bihemispheric | 1.63 | yes |
| ASL-009 | Left fronto-central [85] | Multifocal bihemispheric | 0.76 | yes |
| ASL-010 | GTC, Bifrontal, maximum left [166] | Left frontal | 3.82 | yes |
| ASL-011 | Left fronto-temporal [56] | No Change | 1.10 | n/a |
| ASL-012 | Bitemporal right>left [30] | Multiple areas over right temporal posterior and superior to resection cavity | 2.19 | yes |

of this specificity in frontal and temporal lobe epilepsy. In all eight cases where hypoperfusion >10 mL/100 g/min was observed, we noted that the area of maximal hypoperfusion overlapped with a brain structure subjected to seizure activity (*Table 2*). Thus, hypoperfusion is specifically localized to those areas involved in the seizure and is indeed a local phenomenon.

## Postictal severe hypoxia is mediated by COX-2 activity during seizures

COX-2 is expressed throughout the rat brain (*Figure 4A*) with particularly dense expression in neurons of the hippocampus (*Figure 4B–D*). This expression profile and the well-established role in neurovascular coupling (*Niwa et al., 2000*; *Lecrux et al., 2011*; *Lacroix et al., 2015*) made COX-2 a likely candidate responsible for postictal hypoperfusion/hypoxia. We also hypothesized that COX-1 would not play a role since it is involved in the tonic control of brain blood flow through astrocyte signaling (*Rosenegger et al., 2015*). Selective inhibitors for these two main isoforms of COX have been developed and we pre-administered celecoxib or SC-560 to inhibit COX-2 and COX-1, respectively. Neither drug had any significant effect on afterdischarge duration and as predicted, celecoxib (*Figure 4E,F*), but not SC-560 (*Figure 4—figure supplement 1A,B*), prevented postictal severe hypoxia.

We reasoned that a non-selective COX antagonist should also prevent postictal hypoxia. Indeed, pre-administration of acetaminophen dose-dependently inhibited severe hypoxia without altering afterdischarge duration (*Figure 4G,H*). Unlike nifedipine, administering acetaminophen immediately following termination of the afterdischarge had no effect on the postictal hypoxic profile (*Figure 4—figure supplement 1C,D*). Since acetaminophen had to be present at the time of the afterdischarge, this suggests that COX-2 activity *during* the afterdischarge gives rise to prolonged postictal severe hypoxia.

We then returned to the in vitro preparation to directly observe the effect of COX inhibition on vasoconstriction. Slices were treated with acetaminophen (100 μM) or vehicle (0.1% DMSO) and then stimulated for 2 min at 3 Hz to drive epileptiform-like activity. Indeed, acetaminophen

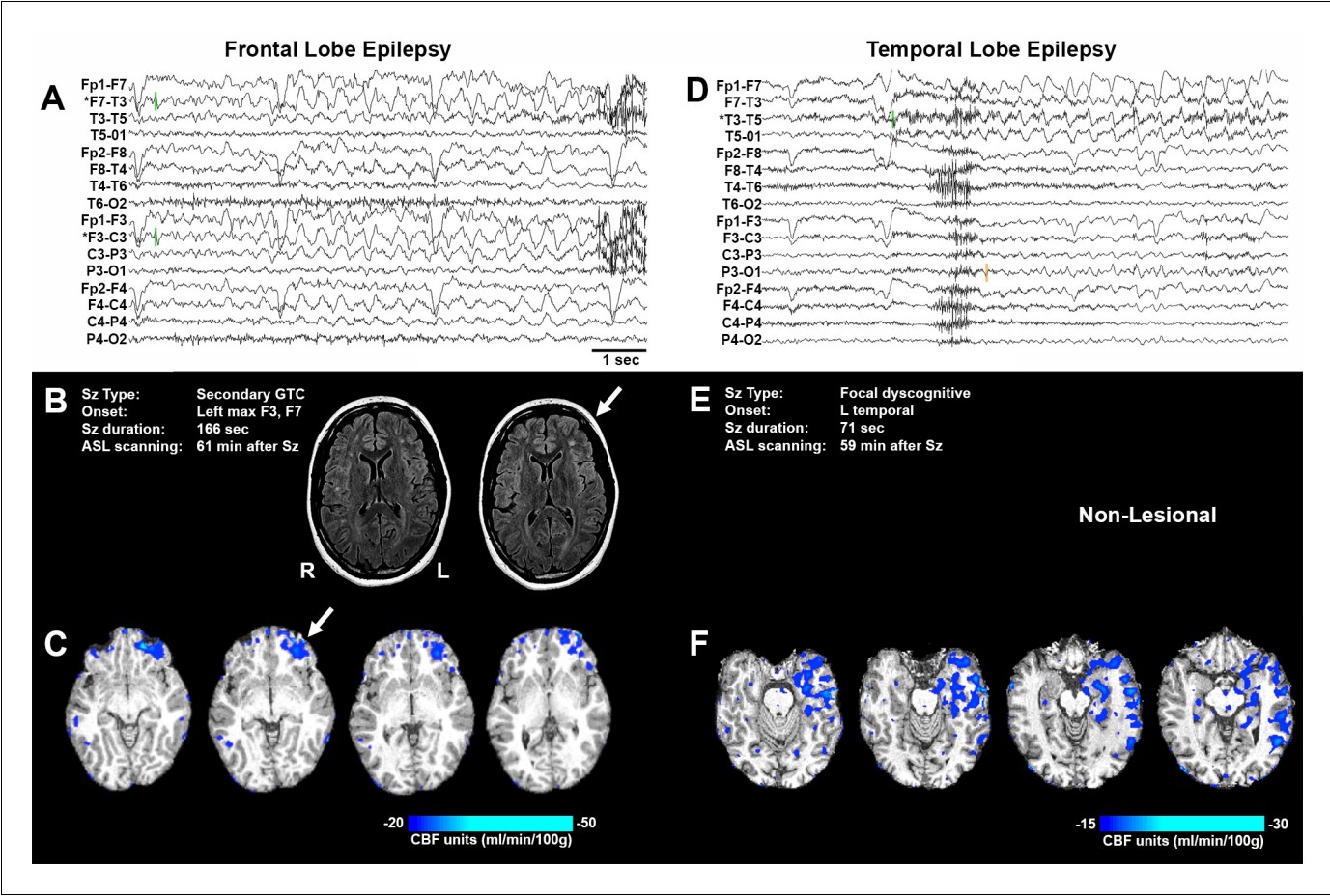

**Figure 3.** Representative seizure-specific local postictal hypoperfusion in clinical epilepsy. (**A**) ASL-010, 26 year old female with drug resistant focal epilepsy. Ictal EEG recording on a longitudinal bipolar montage with left frontal seizure onset and spread to the right frontal region. Green bars indicate seizure onset. * indicates the electrodes to localize seizure onset. Scale bar = 1 s. and also applies to (**D**). (**B**) Seizure description and MR. Fluid-attenuated Inversion recovery (FLAIR) MR images demonstrating area of poor gray-white matter differentiation with subcortical white matter hyperintensity over the left frontal region (arrows). R and L indicate right and left sides and also apply to (**C**) and (**F**). (**C**) Subtraction CBF map (inter-ictal – post-ictal) superimposed onto the patient's T1-weighted anatomical image indicating areas of left frontal hypoperfusion > 20 mL/100 g/min (> 30% reduction compared normal gray matter CBF). (**D**) ASL-001, 38 year old right handed male with intractable non-lesional epilepsy. Ictal EEG recording on a longitudinal bipolar montage with left temporal seizure onset (green bar) and spread to the left parasagittal region (orange bar). (**E**) Seizure description. Patient was non-lesional. (**F**) Subtraction CBF map shows profound hypoperfusion (>15 mL/100 g/min) in left temporal lobe.

The following figure supplement is available for figure 3:

**Figure supplement 1.** Clinical postictal hypoperfusion is more severe with longer seizures.

prevented vessel constriction following stimulation (*Figure 4I*), but unlike nifedipine (*Figure 2K*), had no effect on vessel diameter prior to stimulation. This mirrors our data obtained in vivo. Taken together, these data support our hypothesis that COX-2 activity during a seizure ultimately leads to sustained vessel constriction and severe hypoxia.

To further test the role of COX-2 in the generation of postictal severe hypoxia, we elicited afterdischarges in mice with a genetic knockdown of COX-2 function (*Figure 5A*). Specifically, these mice (PTGS2$^{Y385F}$) have a single amino acid substitution in cyclooxygenase active site, have intact peroxidase function, and have blunted prostanoid production in response to LPS (*Yu et al., 2006*). Both mutant and C57BL/6J controls exhibit brief afterdischarges, which did not differ in duration in response to electrical kindling stimulation. However, only C57BL/6J controls showed severe hypoxia during the postictal period, while the mutants remained within normoxia (*Figure 5B,C*).

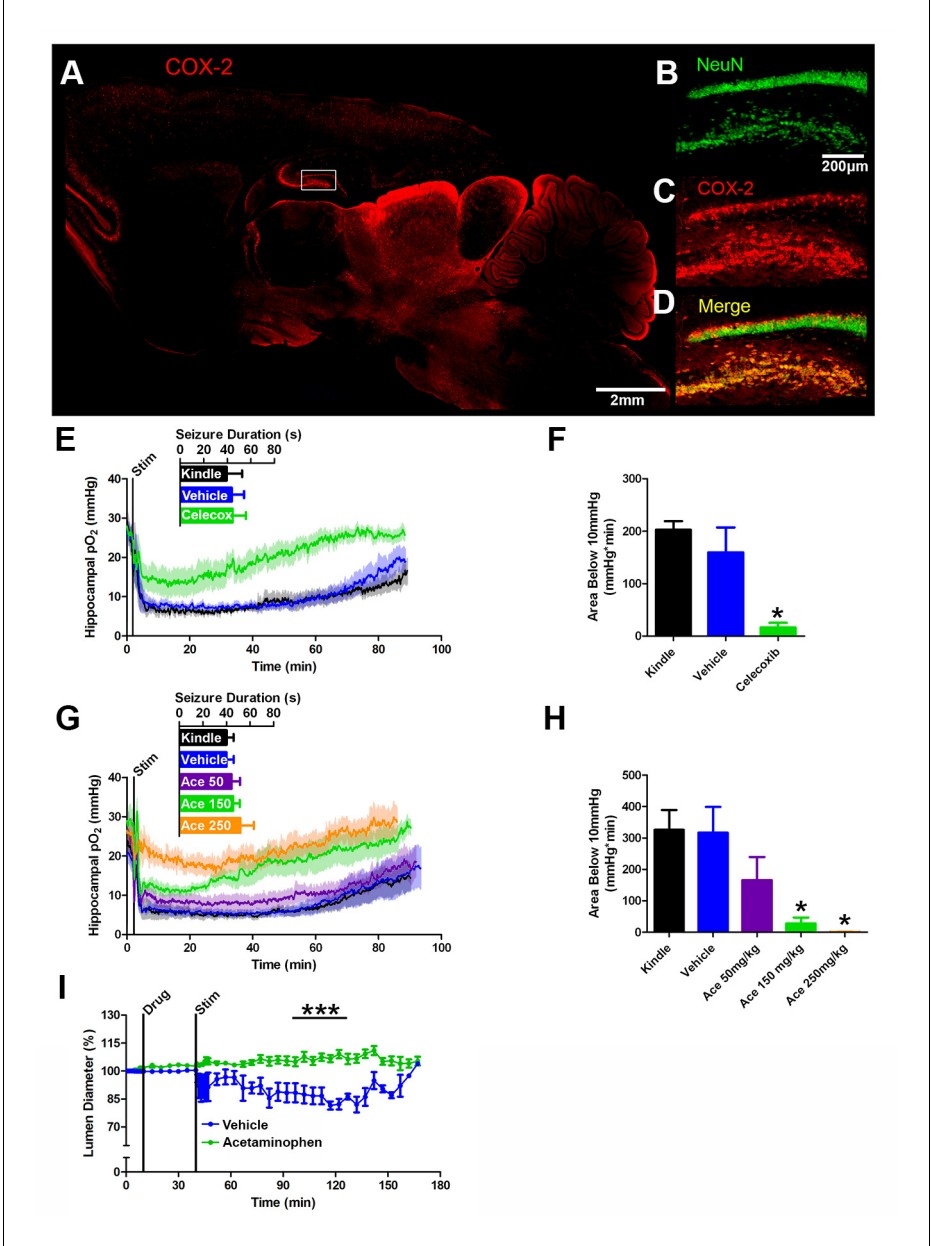

**Figure 4.** COX-2 activity during a seizure is required for postical severe hypoxia. (**A**) COX-2 expression in sagittal section. Hippocampal inset is displayed (see **B–D**). Scale bar = 2 mm. (**B–D**) Contents of inset from (**A**). Many NeuN-expressing neurons (**B**) in DG and hilus express COX-2 (**C**). (**D**) displays colocalization of NeuN and COX-2. Scale bar = 200 μm. (**E**) Celecoxib pre-treatment (20 mg/kg) inhibited severe hypoxia during the postical period (n = 5). Inset reveals no difference in seizure duration. (**F**) Celecoxib caused a significant reduction in the area below 10mmHg. *p<0.05 (within subject ANOVA). (**G**) Acetaminophen (Ace) dose-dependently inhibited postical severe hypoxia (n = 5). Doses are listed on inset (50–250 mg/kg). Inset reveals no difference in seizure duration. (**H**) 150 mg/kg and 250 mg/kg of acetaminophen significantly decreased the area below 10mmHg. *p<0.05 (within subject ANOVA). (**I**) Lumen diameter over time following application of acetaminophen (100 μM; n = 7 or vehicle (n = 5). Acetaminophen treatment prevented post-stimulation constriction (analyzed between 60–90 min post-stim). ***p<0.001.

The following figure supplement is available for figure 4:

**Figure supplement 1.** COX-1 or postical COX-1/2 inhibition do not inhibit severe hypoxia.

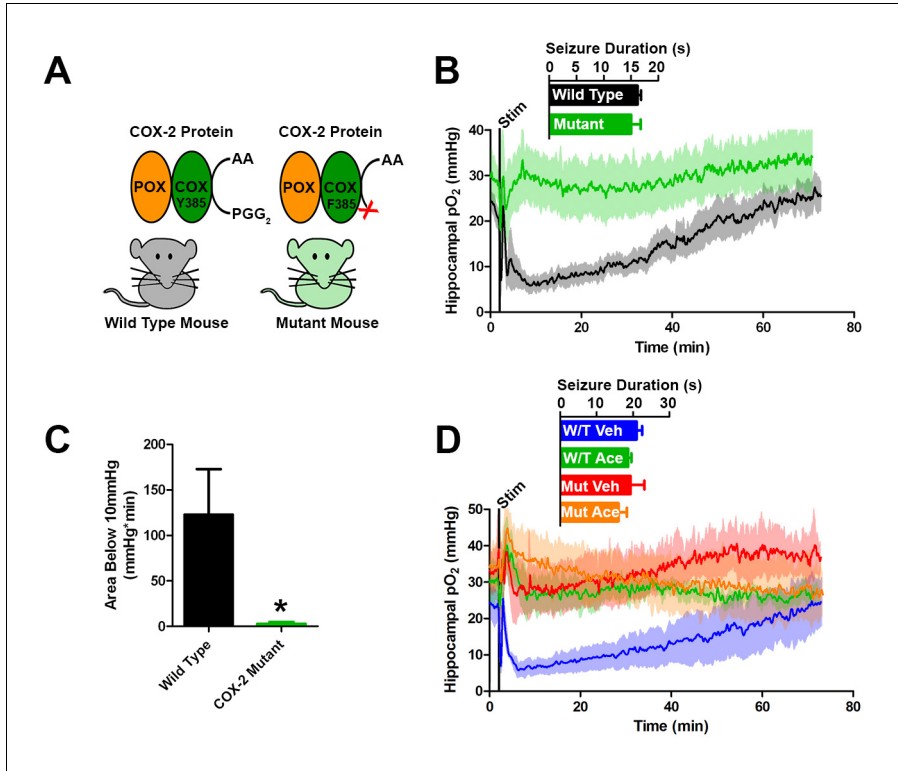

**Figure 5.** Genetic knockdown of COX-2 function prevents postictal severe hypoxia. (**A**) COX-2 proteins from wild-type and mutant mice (PTGS2$^{Y385F}$) displayed as a schematic. Both mice have functional peroxidase (POX). Mutants have a point mutation in the cyclooxygenase (COX) active site, which prevents the conversion of arachadonic acid (AA) to prostaglandin $G_2$ (PGG$_2$). (**B**) No differences were observed in seizure duration between wild-type and mutant mice (n = 4). Wild-type mice displayed severe postictal hypoxia, while mutant mice remained normoxic in the postictal period. (**C**) Quantification of (**D**). COX-2 mutant mice displayed a significant reduction in severe hypoxia (area below 10 mmHg). *p<0.05 (t-test). (**E**) Wild-type and mutant mice were treated with acetaminophen (250 mg/kg) (n = 3). Acetaminophen prevented hypoxia in wild-type mice and had no effect in mutant mice.

Furthermore, severe hypoxia was blocked by acetaminophen pre-treatment in C57BL/6J controls corroborating our results in rats and no further effect was seen in COX-2 mutants (*Figure 5D*).

## Pharmacological profile of postictal severe hypoxia

To better understand how seizures lead to this prolonged period of hypoperfusion/hypoxia we tested several pharmacological agents with diverse mechanisms of action with an eye to identify other potential drug candidates that might prevent hypoperfusion/hypoxia. Data are summarized in *Table 3*.

We further interrogated the COX-2 pathway by pre-administering, ibuprofen, which non-selectively inhibits both COX-1 and COX-2. Likewise, we also co-administered celecoxib and SC-560 to specifically and concurrently inhibit both enzymes. With both methods, inhibiting COX-1 and-2 achieved greater than 95% inhibition of severe hypoxia, similar to acetaminophen. This is of interest since COX-2 inhibition alone achieved a 67.26 ± 27.42% reduction in the severity of hypoxia and SC-560 had no significant effect. These results support a role for an interaction between COX-1 and COX-2, which has been demonstrated through COX-1/COX-2 heterodimer signaling (*Yu et al., 2006*).

The conversion of arachidonic acid to PGH2 by COX-2 is followed by PGE2, PGD2, PGF2, PGI2, and TXA2 production by specific synthases (*Hla and Neilson, 1992*) and subsequent constriction or dilation though G-protein coupled receptor signaling. Specific inhibitors for each of these pathways are not available, but we were able to target two of them: PGE2 and TXA2. We inhibited PGE2

**Table 3.** Investigation of mechanisms involved in postictal severe hypoxia.

| Drug | Principle known mechanism of action | $\Delta$ severity of hypoxia $\frac{\text{(Veh−Drug)}}{\text{Veh}} \times 100\%$ |
|---|---|---|
| Nifedipine (15 mg/kg) | L-type $Ca^{2+}$ Channel Blocker | +78.82 ± 7.632% *** |
| Nifedipine (15 mg/kg postictal) | | +87.33 ± 10.49% ** |
| Acetaminophen (250 mg/kg) | COX-1/2 Inhibitor | +99.88 ± 0.12% *** |
| Acetaminophen (150 mg/kg) | | +88.88 ± 7.355% *** |
| Acetaminophen (50 mg/kg) | | +48.85 ± 26.61% |
| Acetaminophen (250 mg/kg postictal) | | −3.33 ± 17.29% |
| Ibuprofen (20 mg/kg) | | +99.23 ± 0.396% *** |
| Celecoxib (20 mg/kg) | COX-2 Inhibitor | +67.26 ± 27.42% * |
| SC-560 (20 mg/kg) | COX-1 Inhibitor | +19.42 ± 15.37% |
| Celecoxib (20 mg/kg) + SC-560 (20 mg/kg) | COX-2 and COX-1 Inhibitors | +95.80 ± 4.21% *** |
| Acetaminophen (250 mg/kg) + Nifedipine (15 mg/kg) | COX-1/2 Inhibitor + L-type $Ca^{2+}$ Channel Blocker | +100 ± 0.00% *** |
| CAY-10526 (2 mg/kg) | Prostaglandin E2 Synthesis Inhibitor | +41.72 ± 9.94% ** |
| Seratrodast (10 mg/kg) | Thromboxane A2 Receptor Antagonist | −12.91 ± 44.08% |
| Ozagrel (10 mg/kg) | Thromboxane A2 Synthesis Inhibitor | +21.24 ± 16.42% |
| 2-APB (3 mg/kg) | IP3r Antagonist + TRP Channel Blocker | +45.27 ± 18.44% * |
| Chelerythrine Chloride (15 mg/kg) | PKC Inhibitor | +25.54 ± 29.90% |
| Milrinone (3 mg/kg) | Phosphodiesterase-3 Inhibitor | −10.74 ± 34.31% |
| Sildenafil (15 mg/kg) | Phosphodiesterase-5 Inhibitor | −19.34 ± 14.03% |
| SKA-31 (10 mg/kg) | $IK_{Ca}$ Channel Activator | −1.50 ± 12.68% |
| Paxilline (2.5 mg/kg) | $BK_{Ca}$ Channel Blocker | +19.55 ± 14.98% |
| L-Arginine (500 mg/kg) | Nitric Oxide Precursor | −5.05 ± 14.08% |
| Fasudil (10 mg/kg) | Rho Kinase Inhibitor | −53.07 ± 17.27% * |

All drugs were delivered by intraperitoneal injection pre-seizure (unless otherwise stated).

Statistics reported as different from chance (one sample T-test).

*p < 0.05, **p<0.01, ***p<0.001.

+ number indicates inhibition of hypoxia.

− number indicates potentiation of hypoxia.

synthesis with CAY-10526, which resulted in a + 41.72 ± 9.94% reduction in the area below 10 mmHg. Disruption of TXA2 signaling by inhibiting its synthesis (ozagrel) or receptor (seratrodast) resulted in no significant change from vehicle. Thus, part of the COX-2 signaling pathway can be explained by PGE2 production.

Since, nifedipine inhibits calcium entry into vascular smooth muscle, we postulated that inhibiting the increase in free calcium with 2-APB (primarily through IP3 receptors and TRP channels) should also inhibit postictal hypoxia. Indeed, we observed a 45.27 ± 18.44% decrease in the area below 10 mmHg with 2-APB pre-treatment. We then hypothesized that the rise in free calcium could activate protein kinase C (PKC), which can sustain vasoconstriction by inhibiting myosin light chain phosphatase through CPI-17 (*Kitazawa et al., 2003*). However, PKC inhibition was unable to prevent severe postictal hypoxia. Furthermore, activating vasodilatory pathways by increasing cAMP (milrinone), (*Farrar, 1991*) increasing cGMP (sildenafil), (*van den Brink et al., 2000*) hyperpolarizing smooth muscle with $IK_{Ca}$ activator (SKA-31), (*Maloney-Wilensky et al., 2009*) preventing astrocytic release of potassium onto smooth muscle via $BK_{Ca}$ (paxilline), or (*Leung et al., 2000*) increasing production of nitric oxide (L-arginine) provided no inhibition of postictal hypoxia. In fact, Rho Kinase inhibition with fasudil achieved the opposite effect and augmented postictal severe hypoxia perhaps by distributing blood flow to areas outside of the afterdischarge zone. Together, these data provide support for the inflexible nature of postictal hypoperfusion/hypoxia, which is not amenable to several

vasodilatory treatments, and highlights the central roles COX-2 activation and the sustained calcium elevation in vascular smooth muscle.

## Anti-seizure drugs (ASDs) do not inhibit postictal severe hypoxia

Despite taking ASDs, many persons with epilepsy still experience breakthrough seizures and 30–40% are completely refractory to ASD treatment. Given the potential negative consequences of prolonged periods of severe local tissue hypoxia, an important question for us was whether the commonly prescribed ASDs modulate this phenomenon. Fortunately, using electrical kindling stimulation well above seizure threshold did not affect seizure duration following ASD treatment and allowed us to observe the effect of ASDs on postictal severe hypoxia without that confound. Acute dosing with commonly prescribed ASDs had no significant effect on postictal severe hypoxia when compared to vehicle treatment (*Table 4*). Interestingly, ethosuximide reduced the severity of hypoxia by 47.61 ± 11.01%. Ethosuximide is used to treat absence seizures by inhibiting T-type calcium channels, but partial inhibition of L-type calcium channels has been reported (*Kostyuk et al., 1992*).

## Blocking postictal severe hypoxia prevents postictal behavioral disruption

In order to determine the role of postictal severe hypoxia in postictal behavioral disruption we first began by addressing Todd's paresis as it was originally described: motor weakness following seizures that involve motor cortex (*Todd, 1849*). We hypothesized that the postictal hypoperfusion/hypoxic period would be responsible for motor impairment and that this impairment would also be confined to the hypoperfusion/hypoxic period. To test this, we used nifedipine as a tool to prevent postictal hypoperfusion/hypoxia, allowing us to dissociate the effects of an afterdischarge from the combined effects of an afterdischarge with hypoperfusion/hypoxia on behavior. We targeted the corpus callosum at the level of the forelimb area of motor cortex (*Teskey et al., 2002*) as the site to elicit bilateral afterdischarges. The hanging bar test was used to assess motor weakness. We first measured baseline hang time (standardized to 100%) and then elicited an afterdischarge. Hang time was repeatedly measured at 20, 40 and 80 min postictal (*Figure 6A*) whilst neocortical pO$_2$ was continuously recorded throughout. As expected, pretreatment with nifedipine prevented the expression of postictal severe hypoxia without affecting seizure duration (*Figure 6B*). Although severely hypoxic at both the 20- and 40 min time-point, the vehicle-treated seizure group displayed a significantly shorter hang time at 40 min postictal (*Figure 6C*), suggesting that the duration of severe hypoxia is important for the generation of behavioral deficits. At 80 min post-afterdischarge, the non-treated group's pO$_2$ levels returned to normoxia and the behavioral performance was equivalent to

**Table 4.** Effect of Anti-Seizure drugs on postictal severe hypoxia.

| Drug | Principal known mechanism of action | Δ severe hypoxia $\frac{(\text{Veh}-\text{Drug})}{\text{Veh}} \times 100\%$ |
|---|---|---|
| Ethosuximide (300 mg/kg) | T-type Ca$^{2+}$ Channel Blocker | +30.53 ± 19.31% * |
| Topiramate (50 mg/kg) | Na$^+$ Channel Blocker, GABA Enhancement, AMPA Inhibition | +4.34 ± 14.07% |
| Bumetanide (2.5 mg/kg) | NKCC1 Transporter Inhibitor | −0.92 ± 14.72% |
| Phenobarbital (30 mg/kg) | GABA Receptor Agonist | −1.22 ± 13.90% |
| Levetiracetam (250 mg/kg) | Glutamate Release Inhibition | −14.85 ± 19.71% |
| Phenytoin (75 mg/kg) | Na$^+$ Channel Blocker | −15.76 ± 17.00% |
| Lamotrigine (15 mg/kg) | Na$^+$ Channel Blocker | −20.95 ± 17.13% |
| Valproate (150 mg/kg) | Na$^+$ Channel Blocker, GABA Enhancement | −24.89 ± 22.65% |

All drugs were delivered by intraperitoneal injection pre-seizure.

Statistics reported as different from chance (one sample T-test).

*p<0.05.

+ number indicates inhibition of hypoxia.

-number indicates potentiation of hypoxia.

 

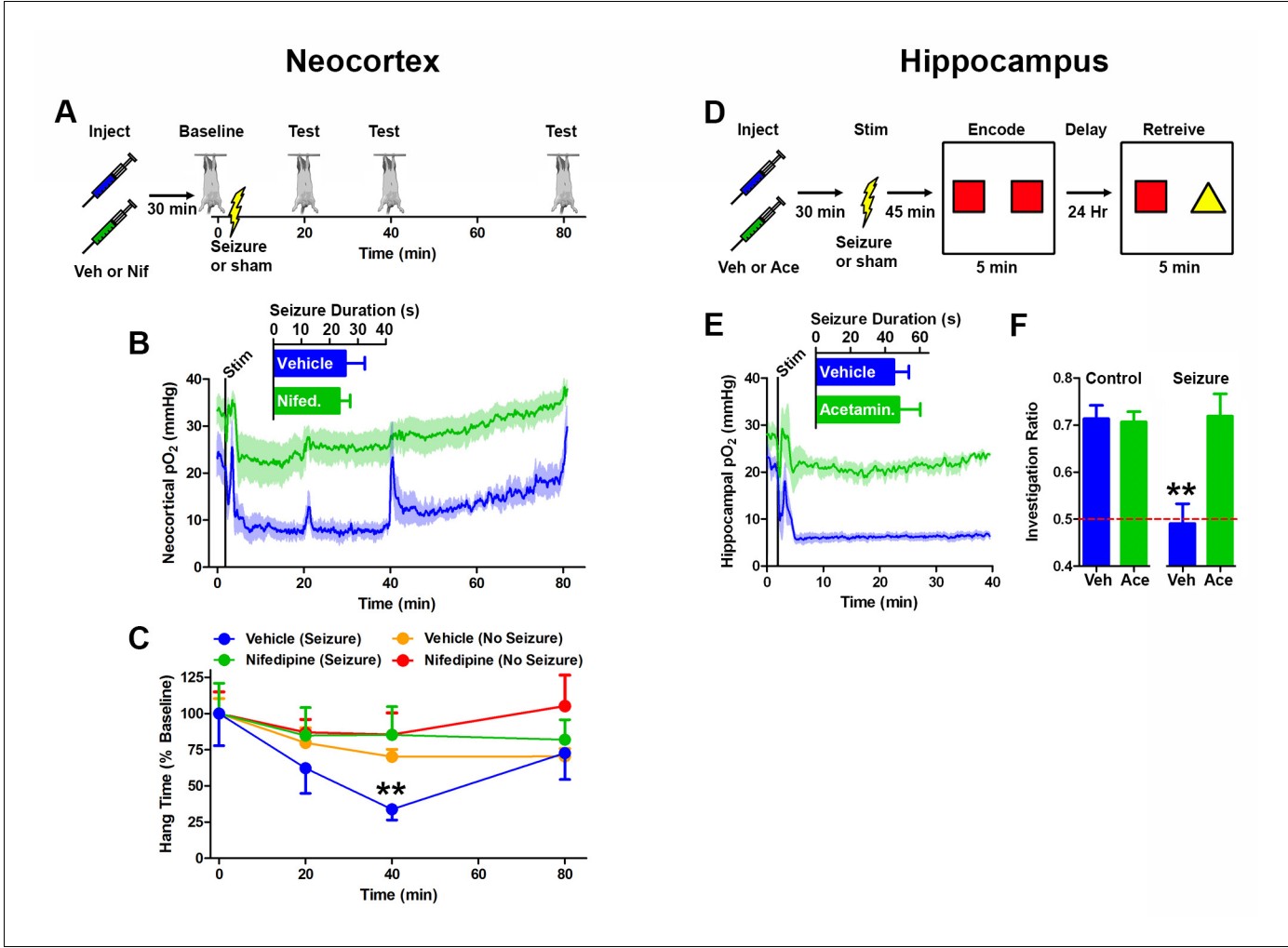

**Figure 6.** Preventing postictal severe hypoxia prevents behavioral impairments. (A) Experimental timeline for neocortex experiments. Rats were injected with vehicle or nifedipine 30 min prior to a baseline hang time measurement. Neocortical seizures (or sham) were then elicited and rats were retested at 20, 40, and 80 min post-seizure. (B) Nifedipine pre-treatment (15 mg/kg) inhibited severe hypoxia during the postictal period (n = 4). Inset reveals no difference in seizure duration. Note that during task performance motor cortex pO2 levels temporarily increased while the rat hung from the bar, likely mediated by increased blood pressure, but returned to pre-test levels following release from the bar. (C) Performance on the hanging bar task was steady across the experiment, except in the vehicle+seizure group (hypoxic) (n = 4 for seizures, n = 3 for no seizures). A significant decrease was observed at 40 min post-seizure, which recovered by 80 min. Data are mean ± SEM. **p<0.01 (2 way ANOVA). (D) Experimental timeline for hippocampus experiments. Rats were injected with vehicle or acetaminophen (250 mg/kg) 30 min prior to hippocampal seizure induction (or sham). 45 min post-seizure, rats were place in testing environment to encode the memory of 2 identical objects. 24 hr later, memory was tested by measuring the percentage of time spent investigating the novel object. (E) No differences were observed in seizure duration (n = 4). Acetaminophen prevented postictal hypoxia. (F) Rats that had seizures and vehicle treatment (hypoxic) performed no different from chance and significantly worse than all other groups (n = 4). Data are mean ± SEM. **p<0.01 (ANOVA).

nifedipine treated and non-seizure groups. These results indicate that it is the postictal severe hypoxic period that is responsible for the motor weakness, not the seizure per se.

To determine whether postictal severe hypoxia was the cause of memory disruption we turned to a hippocampal dependent memory task: novel object recognition. Following habituation to the testing environment, hippocampal afterdischarges were elicited in the presence or absence of acetaminophen and rats were exposed to two identical objects at a time of severe hypoxia (45 min post-discharge). 24 hr later, rats were re-introduced to the testing chamber in the presence of a novel and familiar object (*Figure 6D*). Since rats prefer to explore the novel object more frequently, a preference for the novel object indicates a memory was formed for the familiar object. Rats that were

severely hypoxic at the time of memory encoding (seizure+vehicle, *Figure 6E*) explored the novel and familiar objects with no significant preference 24 hr later, which indicates that they were unable to form the memory during the postictal severe hypoxia (*Figure 6F*). Moreover, the rats that had seizures without severe hypoxia (acetaminophen pre-treatment) performed at the same level as controls (acetaminophen or vehicle without seizures). These results, along with the ability of nifedipine to prevent Todd's paresis, support the hypothesis that severe hypoxia following seizures, and not the seizures themselves, are the underlying cause of these specific postictal impairments.

## Discussion

Using a multi-level approach from animal models to clinical epilepsy, we demonstrate a profound postictal hypoperfusion/hypoxic event. We also demonstrated significant post-ictal hypoperfusion in 8 of 10 patients. The reasons why two subjects did not exhibit significant post-ictal hypoperfusion are unclear, but could relate to the brief duration of their seizures and/or patient heterogeneity. In rodents we identified COX-2 activity during afterdischarges as a central player in mediating this pathophysiological response. Inhibiting COX-2 or restoring perfusion with the potent vasodilator, nifedipine, prevented ensuing severe hypoxia and averted postictal behavioral impairments. Thus, brief seizures initiate a local and long-lasting hypoperfusion/hypoxic insult, which is responsible for behavioral dysfunction in the postictal period rather than the seizure itself.

Prior to this report the direction of postictal hemodynamic changes was unclear since conflicting reports indicated either postictal hypoperfusion (*Rowe et al., 1991*; *Newton et al., 1992*; *Leonhardt et al., 2005*) or hyperperfusion (*Fong et al., 2000*; *Tatlidil, 2000*; *Hassan et al., 2012*). This discordance is most likely explained by the variable timing of the single postictal scans. In our rat models we observed the most severe hypoxia occurred between 20–60 min post-seizure thus we performed clinical postictal scans in this time window and observed dramatic local hypoperfusion. Importantly, only those previous studies that imaged during this period observed hypoperfusion, which highlights the importance of timing with this phenomenon (*Rowe et al., 1991*; *Newton et al., 1992*; *Leonhardt et al., 2005*). Moreover, the timing of postictal scans is an important consideration as these techniques may aid the ability of clinicians to locate the seizure onset zone in potential surgical candidates (*Duncan et al., 1993*).

A limitation of this study is the tool used to measure hypoperfusion in the rat. Laser Doppler flowmetry was used for in vivo recordings, which is an indirect measure of blood perfusion. However, we used this system in conjunction with our highly accurate $pO_2$ recordings in the same region of the hippocampus and observed hypoperfusion and hypoxia that display similar kinetics. For a more direct measure of vasoconstriction, we used acute hippocampal slices and observed the diameter of local arterioles. The conditions in slice differ substantially from in vivo conditions and this could potentially alter the results. However, the degree of constriction observed in slice predicts a similar change in blood perfusion to what we observed with laser Doppler flowmetry. Furthermore, the vasoconstriction following seizure stimulation in slice was sensitive to acetaminophen and nifedipine, which supports the notion that this constriction follows the same set of mechanisms that occur in vivo. In conjunction with the clinical observations of hypoperfusion, these data support the central role of hypoperfusion in the generation of severe hypoxia in the postictal period.

Prior to this study, few have investigated local tissue oxygenation during the seizure event itself and they identified a transient 'dip' at seizure onset that quickly recovers (*Bahar et al., 2006*; *Suh et al., 2006*). The longest of these recordings included 30 s of postictal oxygen sampling and took place under ketamine anesthesia, which dramatically reduces neurovascular coupling by inhibiting NMDA-dependent COX-2 activation (*Tran and Gordon, 2015*). It is therefore unsurprising that postictal hypoperfusion/hypoxia was not identified in these studies. One group discovered hypoxyprobe-positive neurons following pilocarpine-induced SE and during the period of recurrent spontaneous seizures (three weeks post SE), which they attributed to the hypoxic dip that occurs during seizures (*Gualtieri et al., 2013*). We also observed a transient dip in oxygenation at seizure onset, but the duration of this dip is dwarfed by the prolonged period of severe hypoxia during the postictal period and likely accounts for hypoxyprobe labeling.

We identified that COX-2 plays a central role in coordinating a cascade of events to induce severe hypoxia following electrographic seizures. Activity-dependent induction of COX-2 in neurons plays an important role in normal neurovascular coupling by producing vasoactive prostanoids that act on

blood vessel receptors to cause vasodilation (*Niwa et al., 2000*; *Lecrux et al., 2011*; *Lacroix et al., 2015*). Under pathophysiological activation during electrographic seizure activity, we observed that vasodilation is quickly switched to prolonged vasoconstriction. We also found that inhibiting PGE2 synthesis provided partial inhibition of severe hypoxia. The EP1 and EP3 receptor subtypes for PGE2 have been observed to induce vasoconstriction (*Jadhav et al., 2004*) and may mediate part of this response.

Our study also investigated several other pathways by which postictal severe hypoxia could be mediated and also aimed to identify other drug candidates to inhibit this phenomenon. Of particular note was nifedipine's inhibition of severe hypoxia even when administered after the seizure. Furthermore, this strategy is important clinically since administering a drug after a seizure avoids the challenges of seizure prediction. We also tested commonly prescribed anti-seizure medications since it would be informative to know which of these medications potentially inhibit, or more importantly, exacerbate postictal severe hypoxia in patient populations. Most drugs, including anti-seizure medications, failed to modulate this postictal severe hypoxia, which we believe emphasizes that this is a pathological phenomenon and that breakthrough seizures while on antiseizure medications do not directly protect against this insult.

Postictal behavioral impairments last much longer than the seizures themselves and negatively impact quality of life. The mechanism underlying these disruptions were previously unknown and no treatment options existed. Our data clearly demonstrates that two common postictal impairments, Todd's paresis and amnesia, can be prevented by interfering with the pathways that lead to hypoperfusion/hypoxia. Thus, L-type calcium antagonists and COX-2 inhibitors are strong candidates for the treatment of these currently untreated behavioral impairments.

Our discovery that brief and mild seizures lead to an extended hypoperfusion/hypoxic event is critically important because it establishes that seizures could injure the brain through *postictal severe hypoxia*, and not necessarily by the seizure itself. Severe hypoxia may be an important component of seizure-induced brain damage (*Jackson et al., 1999*) and since postictal hypoperfusion/hypoxia is COX-2 dependent, this hypothesis can be assessed. Though there is some controversy (*Rojas et al., 2014*), it is generally accepted that COX-2 inhibition is neuroprotective from seizures. Indeed, with genetic or pharmacological COX-2 inhibition, seizure-induced brain damage can be dramatically reduced (*Serrano et al., 2011*; *Kunz and Oliw, 2001*; *Hewett et al., 2006*). Though at the time of these studies researchers were unaware of postictal hypoperfusion/hypoxia and the requirement of COX-2 in this response, they provided support for the hypothesis that postictal hypoperfusion/hypoxia injures the brain. Given the central role of brain injury in epileptogenesis (*Sloviter and Bumanglag, 2013*), preventing injury from hypoperfusion/hypoxia may be a novel preventative treatment strategy in epilepsy.

## Materials and methods

### Animals

Young adult male Hooded Long-Evans (LE) rats weighing between 250–300 g at the start of experimentation were used in this study (Charles River, Canada). Young adult male C57BL6/J and PTGS2$^{Y385F}$ mice weighing between 21–30 g were also used (Jackson Laboratory, RRID: IMSR_JAX: 008101). Rats and mice were housed individually in clear plastic cages and were maintained on a 12:12 hr light/dark cycle lights on at 07:00 hr, in separate colony rooms under specified pathogen free conditions. Food and water were available *ad libitum*. All experimental procedures occurred during the light phase.

### Eliciting and recording seizures (kindling) and oxygen detection

Electrodes were constructed from Teflon-coated, stainless steel wire, 178 µm in diameter (A-M Systems, Sequim, WA). Wire ends were stripped of Teflon and connected to gold-plated male amphenol pins. Rats were anaesthetized with a 5% isoflurane, and maintained between 1% and 2%. Lidocaine (2%) was administered subcutaneously at the incision site. One bipolar electrode was chronically implanted under stereotaxic control (*Paxinos and Watson, 1986*) in the site of interest with an oxygen-sensing probe positioned nearby (*Table 5*). The implants were adhered and anchored to the skull using dental cement and six stainless steel screws. One of the six screws served

**Table 5.** Stereotaxic coordinates for surgical implantation.

| Experiment | Anterior(+)/ Posterior(−) | Lateral Right(+)/Left(−) | Ventral (from brain surface) |
|---|---|---|---|
| Rat Dorsal Hippocampus | Electrode: −3.0 mm<br>Optode: −3.5 mm | Electrode: 0.5 mm<br>Optode: 3.5 mm | Electrode: 3.5 mm<br>Optode: 3.5 mm |
| Rat Dorsal Hippocampus with LDF | Electrode: −3.0 mm<br>Optode: −5.0 mm<br>LDF probe: −3.0 mm | Electrode: 0.5 mm<br>Optode: 2.2 mm<br>LDF probe: 3.5 mm | Electrode: 3.5 mm<br>Optode: 3.5 mm<br>LDF probe: 3.5 mm |
| Rat Ventral Hippocampus | Electrode: −4.5 mm<br>Optode: −3.0 mm | Electrode: 4.5 mm<br>Optode: 3.5 mm | Electrode: 6.5 mm<br>Optode: 3.5 mm |
| Rat Ventral Hippocampus w/ Cannula | Electrode: −4.5 mm<br>Optode: −3.0 mm<br>Cannula: - 5.8 mm | Electrode: 4.5 mm<br>Optode: 3.5 mm<br>Cannula: −4.5 mm | Electrode: 6.5 mm<br>Optode: 3.5 mm<br>Cannula: 4.3 mm |
| Rat Neocortex | Electrode:+1.0 mm<br>Optode: 0 mm | Electrode: 0.5 mm<br>Optode: 3.0 mm | Electrode: 3.6 mm<br>Optode: 1.5 mm |
| Mouse Ventral Hippocampus | Electrode: −2.9 mm<br>Optode: −1.6 mm | Electrode: 3.0 mm<br>Optode: 2.0 mm | Electrode: 3.0 mm<br>Optode: 1.8 mm |

as a ground electrode. Subsequent experimental procedures commenced no earlier than five days following surgery.

Oxygen recordings were obtained using an implantable fiber-optic oxygen-sensing device. 525 nm light pulses induce fluorescence (measured at 650 nm) at the platinum tip that is quenched by oxygen within a local area (~500 $\mu m^3$) and uses the fluorescence decay time to derive $pO_2$ (*Ortiz-Prado et al., 2010*). The technology (Oxylite, Oxford Optronics, United Kingdom) does not consume oxygen while measuring absolute $pO_2$ values. The manufacturer individually calibrates each biologically inert probe, called an optode. The implantable system was designed by JFD in collaboration with Oxford Optronics. The implant is inserted under isoflurane anesthesia. We allow at least seven days between implantation and initiation of measurements to ensure that the effects of acute trauma were minimized. $pO_2$ measurements at 1 Hz can then be made at any time by connecting the implant to the Oxylite using an extension fiber optic lead. The probe provides accurate and continuous measurements of local $pO_2$ levels in brain tissue in *awake, freely moving animals* over several weeks. We used this innovative technology to precisely measure oxygen levels before, during and, most importantly, after seizures. Unlike most animal studies, we were able to perform these measurements without the confounding effect of anesthesia.

On test days, rats were connected to the EEG and oxygen-sensing system and allowed 5 min to adjust before any measurements were taken. A seizure was elicited after 100 s of baseline recording using standard kindling stimulation (1 s train of 1 ms pulses at 60 Hz) delivered through a Grass S88 stimulator (Natus Neurology, Warwick, RI) and seizure duration and stage were recorded (*Racine, 1972*). Once the EEG returned to baseline following a seizure, the electrodes were disconnected, but the fiber-optic cable for oxygen-sensing was left attached. The fiber-optic cable was disconnected when $pO_2$ levels returned to baseline.

## Intrahippocampal kainic acid model of TLE

Rats were chronically implanted with an electrode and oxygen probe as described above. In the contralateral hippocampus, a stainless steel guide cannula (Plastics One) was implanted (see *Table 5* for coordinates). 0.4 $\mu$g of kainic acid (Sigma-Aldrich, Canada) in 0.2 $\mu$L of saline was infused into the hippocampus over 4 min (*Rattka et al., 2013*) using a 1.0 $\mu$L syringe (Hamilton Robotics, Reno, NV) and a microsyringe pump (Harvard Apparatus, model 55–2222, Canada). Following infusion, rats displayed status epilepticus and were allowed to recover for one week. Following recovery, rats were connected to the oxygen and EEG acquisition systems to capture spontaneous seizures.

## Maximal electroconvulsive shock (MES)

A hippocampal electrode and optode were implanted into rats as described above followed by a 1-week recovery period. Rats were then connected to the oxygen detection system and saline soaked

ear clips. A suprathreshold MES stimulus was delivered through the ear clips via a GSC 700 shock generator (model E1100DA) (Grason-Sradler, West Concord, MA). Oxygen levels were recorded before and after the delivery of a 0.2 s train of 60 Hz biphasic sine-wave pulses (*Young et al., 2006*). Seizure duration was recorded by observing seizure behavior.

### 3 Hz seizures

Rats with implanted hippocampal electrodes and optodes were re-used for this experiment. Following baseline oxygen recording, the hippocampus was stimulated with 1 ms square-wave pulses at an intensity of 1 mA and frequency of 3 Hz for 2 min. This drove continuous seizure activity for the duration of the stimulus (*Teskey and Racine, 1993*). $pO_2$ was recorded until the rats recovered from hypoxia.

### Arterial spin labeling in persons with epilepsy

Twelve consecutive patients with focal epilepsy admitted to the Seizure Monitoring Unit at the Foothills Medical Centre for continuous scalp video EEG were prospectively recruited. MR images were collected using a 3T scanner with an 8-channel phase-array head coil (GE Discovery MR750, GE Healthcare). MR data were collected within one hour of a habitual seizure. Baseline (interictal) ASL scans were obtained following a seizure-free period of >24 hr. ASL images were collected using a pseudo-continuous ASL sequence ($1.88 \times 1.88$ mm, 28 slices; 5 mm slice-thickness, 2525 ms post-label-delay, spiral acquisition with 1024 points and eight arms). Quantitative CBF maps were generated by our scanner automatically from both ASL scans in (mL/100 gr/min) using the formula:

$$CBF = 6000 \, \frac{\lambda \left(1 - e^{-\frac{ST}{T1T}}\right) e^{\frac{PLD}{T1B}}}{2T1B(s)\left(1 - e^{\frac{LT}{T1}}\right)\varepsilon * NEX} \left(\frac{\triangle M}{SF * PD_{REF}}\right)$$

where T1B and T1T represent blood and tissue T1 values (1.6 s at 3 T) respectively, $\lambda$ is the partial coefficient set to 0.9, $\varepsilon$ is the efficiency and is set to $0.80 \times 0.75$, $\Delta M$ is the difference between tag and no tag images, $PD_{REF}$ is the reference proton density images, NEX is the number of excitations, SF is a scaling factor of 45, and PLD is the post labeling delay.

Subsequently, CBF maps were registered onto the respective patient's T1-weighted high-resolution image using the FLIRT toolbox Version 5.5 from FSL (*Jenkinson et al., 2002*) using 12 degrees-of-freedom. Finally, a subtraction CBF map was generated from these two normalized CBF maps (baseline minus post-ictal) to identify areas with hypoperfusion in patients post-ictally. We looked for hypoperfusion >10 mL/100 g/min in patients (normal grey matter CBF = 60 mL/100 g/min) (*Chen et al., 2012*).

To obtain the quantitative CBF data, a region of interest (ROI) was manually drawn in a blinded fashion around the single brain area exhibiting the maximal post-ictal blood flow reduction as well as adjacent voxels with CBF changes >10 mL/100 g/min. If no clear hypoperfusion was seen, or if scattered areas of hypoperfusion were seen with no dominant area of hypoperfusion, then an ROI was drawn in the region where the captured seizure originated, based on EEG.

Two Epileptologists (PF/SS) reviewed the clinical data for all patients (e.g., scalp EEG, PET, and SPECT) and reached a consensus on location of the presumed seizure onset zone(s) for all of a given patients seizures. In addition, the same two Epileptologists reviewed the EEG recording of the seizure that was used for the ASL study and based on blinded review of the EEG alone, they identified the brain region where that seizure began. For the purpose of comparison, the brain was divided into the following regions: anterior temporal, posterior temporal, superior parietal, inferior parietal, occipital, orbitofrontal, frontopolar, mesial frontal, and lateral frontal. Next, the location of the most prominent postictal ASL changes were compared to brain region where the presumed seizure onset zone was located. Concordance occurred if the area of maximal hypoperfusion also participated in the seizure.

### Hypoxyprobe™ immunohistochemistry

Hypoxyprobe Green Kit was obtained from Hypoxyprobe™ (Burlington, MA) and used to identify cerebral hypoxia (*Noto et al., 2006*). This kit contains pimonidazole and a mouse anti-pimonidazole-fluorescein isothiocyanate (FITC) monoclonal antibody. Pimonidazole was dissolved in phosphate-

buffered saline and injected into the intraperitoneal cavity at a dose of 30 mg/kg 30 min prior to stimulation. Rats were divided into two groups. Group one was implanted without stimulation and Group two received supra-threshold stimulation. Sixty minutes after stimulation, rats were deeply anaesthetized with sodium phenobarbital and perfused with ice-cold phosphate buffered saline and 4% paraformaldehyde. Brains were rapidly removed post-fixed in 4% paraformaldehyde for 24 hr and cryoprotected in 30% sucrose for at least 48 hr. Coronal sections were cut at a thickness of 25 µm and incubated overnight in 1:50 anti-pimonidazole-FITC antibody. Sections were mounted on gelatin-coated slides and cover slipped with Kystalon (Harleco, VWR, Canada). Images were taken with an Olympus BX51 microscope using a GFP filter set and captured with a QImaging QICAM 1394 camera and ImagePro Plus software. Densely stained cell bodies were counted using the cell counter on the ImageJ software (NIH) in the hilus (*Figure 1G,I*) and CA3 (*Figure 3H,J*). One section between 3.6 and 4.0 mm posterior from bregma as analyzed per animal and mean cell counts were determined within each group. Group means were compared with an independent samples t-test.

## COX-2 immunohistochemistry

Naïve Long-Evans rats were perfused and fixed as was performed with hypoxyprobe immunohistochemistry. 40 µm sagittal slices were incubated in rabbit Anti-COX-2 (1:1000, ab15191, Abcam, RRID:AB_2085144) and mouse anti-NeuN (1:1000, MAB377, Millipore, RRID:AB_2298772) antibodies for 24 hr at room temperature. Sections were then stained with Alexa-594 and Alexa-488-conjugated secondary antibodies (Jackson Immuno, West Grove, PA) to visualize COX-2 and NeuN, respectively. Sections were mounted to gelatin coated slides, cover slipped with DPX (Sigma-Aldrich), and imaged with a slide scanner (Olympus, Canada). A representative image including hippocampus was displayed (*Figure 3A–D*).

## Laser doppler flowmetry (LDF)

The stereotaxic coordinates were slightly modified to accommodate the implantable LDF probe (*Table 5*). The LDF probe and oxygen-sensing probe were implanted at angles to sample adjacent and overlapping region of tissue. LDF was used to assess local blood perfusion using the Oxyflo2000 (Oxford Optronix). Implantable probes were cut to a length of 4 mm to match the depth of the oxygen-sensing probe. Once implanted, recordings were performed on awake, freely moving rats, as previously published using other systems (*Hu et al., 2008*). Data collected from this device were assigned an arbitrary unit on a relative scale from 0–5000 Blood Perfusion Units and collected at a rate of 1 Hz. Data were standardized to baseline (pre-seizure), expressed as a percentage of baseline, and plotted with a moving average of 30 data points for noise reduction.

## Pharmacology and anti-seizure drug delivery

Rats implanted with ventral hippocampal electrodes and dorsal hippocampal optodes were used in these studies. Rats were reused for several drug experiments allowing at least two days between drug treatments. Celecoxib, SC-560, ibuprofen, chelerythrine chloride, milrinone, sildenafil, SKA-31, CAY-10526, seratrodast, ozagrel, paxilline, 2-APB, levetiracetam, and topiramate were obtained from Cayman Chemicals (Ann Arbor, MI). Acetaminophen, nifedipine, bumetanide, ethosuximide, phenytoin, and valproic acid were obtained from Sigma-Aldrich. Lamotrigine was obtained from SelleckChem (Houston, TX). Fasudil was obtained from LC laboratories (Woburn, MA) and phenobarbital was obtained from Strathcona Prescription Center (Canada). Lipophilic drugs were dissolved in 100% DMSO, while hydrophilic drugs were dissolved in saline and injected 30 min prior to seizure induction. The seizure duration and severity of hypoxia (area below 10 mmHg) were compared across kindle (seizure without injection), vehicle-, and drug-treated groups using a within-subject ANOVA and follow-up t-test between vehicle- and drug-treated groups.

## Hippocampal slice preparation

All in vivo and in vitro 3 Hz experiments were performed on P25 – P40 Long-Evans rats (male) and obtained from Charles River Laboratories. Hippocampal slices were prepared as previously described (*Zhou et al., 2011*). In brief, rats were initially anesthetized by isoflurane inhalation in air prior to decapitation. Brains were rapidly dissected from the skull and placed in ice-cooled cutting solution containing the following (in mM): 210 sucrose, 10 glucose, 2.5 KCl, 1.02 $NaH_2PO_4$, 26.19

$NaHCO_3$, 10 $MgSO_4$ and 0.5 $CaCl_2$, pH 7.4, bubbled with 95% $O_2$/5% $CO_2$ at 4°C. Transverse hippocampal slices (370 µm) were sectioned with a vibratome (Leica VT1200, Germany) in cutting solution as described above. Slices were transferred and incubated in oxygenated artificial cerebrospinal fluid (ACSF) containing the following (in mM): 124 NaCl, 5KCl, 1.25 $NaH_2PO_4$, 26 $NaHCO_3$, 2 $CaCl_2$, 1.3 $MgCl_2$ and 10 glucose for 1 hr at 33° to 35°C incubator. Slices were then kept at room temperature prior to visualization of hippocampal blood vessels. Visualization of hippocampal blood vessels were performed using differential interference contrast (DIC) microscope. Vascular lumen diameter was measured using transverse hippocampal slices following 2 min of 3 Hz stimulation applied to Schaffer collaterals in rat hippocampus continuously perfused with ACSF. Hippocampal blood vessels were observed for 2 to 2.5 hr to determine changes in lumen diameter. For experiments involving acetaminophen and nifedipine, 100 µM acetaminophen or 50 µM nifedipine was added to the ACSF, respectively. Slices were pre-incubated with acetaminophen or nifedipine via continuous perfusion for 30 min prior to stimulation. Lumen diameter was measured before and after stimulation and percentage changes in diameter were calculated to determine vascular constriction. Sham stimulation and vehicle (0.1% DMSO) were utilized as control experiments.

## Hanging bar test

Rats that received seizures in this study had an electrode implanted into the corpus callosum at the level of the forelimb movement representation and an oxygen-sensing probe in layer five of the motor neocortex (*Table 5*). Rats recovered for at least a week before experimental procedures were initiated. Rats were introduced to and allowed to explore the testing apparatus before data were obtained. They were permitted to escape to the ends of the rod during this period but the escape routes were blocked off during testing. This served to increase motivation since rats will acclimatize to the task. The test was performed as previously described (*Hunter et al., 2000*), but with modifications. The two forepaws were placed on steel bar with a diameter of 5.0 mm that was suspended from a height of 45 cm. Rats were allowed to hang for a maximum of 45 s and then gently removed to reduce their fatigue for subsequent trials. Each test consisted of the mean of three trials. Nifedipine (15 mg/kg) or vehicle was injected into rats on the first day and the other treatment on the following day. Rats were tested immediately before a seizure (or sham) to establish a baseline and then connected to the EEG and oxygen recording system to elicit and record a seizure, using standard kindling stimulation. In the sham condition, rats did not previously undergo surgery. We then tested at 20, 40, and 80 min after a seizure (or sham). Each group was compared to its own baseline using a one-way ANOVA and Dunnett's Multiple Comparison Test to identify a deficit.

## Novel object recognition task

Rats with ventral hippocampal electrodes and dorsal optodes were used in this study (*Table 5*). Rats then went through the novel object recognition task paradigm (*Clark et al., 2000*), with modifications. Following recovery from surgery, rats were habituated to the environment on the morning of day 1. In the afternoon, rats were injected with vehicle or acetaminophen (250 mg/kg i.p.) to permit or prevent hypoxia following a seizure, respectively. After 30 min rats were stimulated with kindling stimulation and both EEG and local tissue oxygenation were sampled. We confirmed that acetaminophen prevented hypoxia in the rats that received it and did not affect seizure duration, suggesting we have a clear dissociation. Rats were disconnected from the data acquisition system at 45 min postictal and transferred to the testing chamber where they were introduced to two identical objects and allowed to explore them for 5 min. Rats were returned to their home cage and brought back to the testing chamber 24 hr later. This time, a familiar object and a novel object (which differed in color, size, and texture) were introduced and the rats were allowed to explore the objects for 5 min. The position of the objects and type of objects were counter-balanced to account for rat's preference not related to memory. If rats adequately form the memory for the two identical objects on day 1, they should spend more time exploring the novel object on day 2. Each session was videotaped and the duration spent investigating each object was measured. Investigation includes orienting the snout within 2 cm of the object, but not sitting on the object.

## Statistics

All statistical analyses were performed using Prism version 5.01 (GraphPad, La Jolla, CA). t-tests were used for experiments with only two groups. ANOVAs were used for experiments with more than two groups and a follow-up Tukey test to identify in which group(s) the significant differences occur. Repeated measures statistics were used for all within subject experiments.

## Study approval

Rodents were handled and maintained according to the Canadian Council for Animal Care guidelines. These procedures were approved by the Life and Environmental Sciences Animal Care and Health Sciences Animal Care Committees at the University of Calgary (AC11-0073). Human experimentation was approved by the University of Calgary's Conjoint Health Research Ethics Board (REB13-0571). All patients (or guardians of patients) provided written informed consent.

# Acknowledgements

The authors would like to acknowledge Bonita Gunning for technical support, Dr. Tom Feasby for providing an important clinical insight and Drs. Roger Thompson, and Grant Gordon for their critical reading of the manuscript.

# Additional information

### Funding

| Funder | Grant reference number | Author |
|---|---|---|
| Canadian Institutes of Health Research | MOP-136839 | Paolo Federico |
| Canadian Institutes of Health Research | MOP-130495 | G Campbell Teskey |
| Natural Sciences and Engineering Research Council of Canada | RGPIN/03819-2014 | G Campbell Teskey |

The funders had no role in study design, data collection and interpretation, or the decision to submit the work for publication.

### Author contributions

JSF, Conception and design, Acquisition of data, Analysis and interpretation of data, Drafting or revising the article; IG-V, Conception and design, Acquisition of data, Analysis and interpretation of data; MDW, LSD, HID, BLG, XRW, SS, SCS, Acquisition of data, Analysis and interpretation of data; JFD, Conception and design, Analysis and interpretation of data, Drafting or revising the article; MCA, Acquisition of data, Analysis and interpretation of data, Drafting or revising the article; PF, Contributed fMRI experiments, Conception and design, Acquisition of data, Analysis and interpretation of data, Drafting or revising the article; GCT, Discovered the phenomenon in my laboratory and this is the first report of the phenomenon, Conception and design, Analysis and interpretation of data, Drafting or revising the article

### Ethics

Human subjects: Human experimentation was approved by the University of Calgary's Conjoint Health Research Ethics Board (REB13-0571). All patients (or guardians of patients) provided written informed consent.
Animal experimentation: Rodents were handled and maintained according to the Canadian Council for Animal Care guidelines. These procedures were approved by the Life and Environmental Sciences Animal Care and Health Sciences Animal Care Committees at the University of Calgary (AC11-0073).

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
