## [Decision Letter]

Thank you for submitting your article "Postictal behavioural impairments are due to a severe prolonged hypoperfusion/hypoxia event that is COX-2 dependent" for consideration by *eLife*. Your article has been reviewed by three peer reviewers, one of whom, Jan-Marino Ramirez (Reviewer #1), is a member of our Board of Reviewing Editors, and the evaluation has been overseen by Gary Westbrook as the Senior Editor. The following individuals involved in review of your submission have agreed to reveal their identity: Franck Kalume (Reviewer #3). The reviewers have discussed the reviews with one another and the Reviewing Editor has drafted this decision to help you prepare a revised submission.

Summary:

This study investigates the mechanisms underlying behavioral impairments during and in the aftermath of seizures in an animal model and human patients. Using EEG, focal oxygen recordings, and laser Doppler flowmetry, the authors demonstrate that the evoked and spontaneous seizures caused a dramatic reduction in local tissue oxygenation and blood perfusion in three different models of seizures: rat model kindling, maximum electroshock, and 3Hz electrical stimulation seizures. The study is significant since it seems to resolve a significant uncertainty in the field. Previous data have implicated hypoxia in seizure mechanisms, but the findings were in human and thus do not benefit from the better physiological control and mechanistic studies as presented here in the animal model. Moreover, there are also previous human studies that do not observe post-seizure hypoxia, perhaps due to the variable nature of the seizure phenomenon. The study is technically elegant since the authors use sensitive and complementary measures of brain blood flow, which indicate severe ischemia following seizure. The authors compare the animal studies with studies of human patients and demonstrate similar hypoxic and hypoperfusion events in a clinical study of patients with epilepsy. Moreover, employing pharmacological experiments, the investigators demonstrate that the blockade of cyclooxygenase-2 or L-type calcium channels alleviates the seizure-induced hypoxia, hypoperfusion, as well as the resulting post-ictal behavioral impairments. The study suggests that it is possible to separately treat the seizure event and the hypoxic event. This is a clinically significant discovery and the fact that animal studies were combined with the human studies makes this an interesting paper for a general readership.

Essential revisions:

1) The authors throughout the paper define ischemia as blood flow that leads to less than 10mm of mercury partial pressure of oxygen. This measure results in a threshold change for defining hypoxia. This measurement is potentially misleading because many of the treatments are only partially protective and seizures still produce a reduction in PO2, but not all the way to 10mm Hg partial pressure of O2. To make the manuscript more balanced, the authors should remove this thresholding of hypoxia at 10mm of mercury. In cases where area under the curve is presented, the entire area under the curve should be considered not just whether there is a component, which is below 10mm of mercury. Importantly, the authors were able to perform their studies without the confounding effects of anesthesia, which greatly increases the worth of the study.

2) For Figure 2 showing the nifedipine treatment, please calculate panel D based on the entire area under the curve. Furthermore, all statistics in the paper should be based on the total area under the curve and not that below 10mm.

3) Simultaneous blood flow and pO2 are shown in Figure 2, this is important as it suggests that the changes in pO2 are secondary to a reduction in blood flow. However, the sample size of 3 for this key measure seems small. We suggest to add more n's and ideally the effect of the various pharmacological manipulations should be extended to blood flow to confirm that these treatments are not just affecting the utilization of O2 by tissue. We would like to emphasize that these experiments should be relatively straight-forward and manageable within a 2 month time window. If the authors cannot perform these experiments, this should be discussed as a limitation in the Discussion section.

4) The data in Figure 2 showing the effects of repeated stimulation on vessel diameter in slices are interesting. However, these effects could potentially be caused by a mechanistically different effect, because the slices have very different conditions than the in vivo conditions. Note, that in this case the slices are well-stimulated, but not necessarily ischemic. Moreover, the authors are also unable to determine the identity of the vessels within the slice they are examining. Are these arterioles or veins? Thus, the reviewers are not convinced that the authors can conclude a vasoconstriction mechanism without also adding an in vivo measure (2-photon). The authors should discuss this issue as a caveat of the in vitro experiments.

5) The human data in Figure 3 is interesting, although much of the quantification of these data is in a more descriptive form in the tables. We suggest that the authors present these data as mean levels of blood flow in the different groups before and after seizure. The presentation of the human clinical data is done so in a qualitative manner in the tables. This data should be properly quantified in some form of absolute blood flow units that are comparable to the mm of 02 for the animal work. Please also add some information about how the analysis was done. Were investigators blinded to conditions? What exactly are the controls? Potentially, periods without seizures in the same patients and blood flow at this time could be used as a control. Furthermore, given that some of the patients have multiple seizures, it would also be interesting to know whether the results were repeatable on different imaging sessions.

6) "Significant (> 25 CBF units) postictal hypoperfusion was seen in 7 of 10 subjects (Table 2). Two examples are shown in Figure 3. The location of postictal hypoperfusion was concordant with the clinically determined seizure onset zone in four cases and it was in the same lobe of the brain as the seizure onset in two cases. " Please clarify what is meant by significant hypoperfusion > 25 CBF units? Does this mean a change in perfusion of 25 units? Please explain why >25 CBF units would denote hypoperfusion if we don't know what the upper end of the range could be? The correlation with the clinical data does not seem very strong (2 or 4 of 10 examples?). How does this compare to the O2 values from the animal work? Please add this into the Discussion section.

[Editors' note: further revisions were requested prior to acceptance, as described below.]

Thank you for resubmitting your work entitled "Postictal behavioural impairments are due to a severe prolonged hypoperfusion/hypoxia event that is COX-2 dependent" for further consideration at *eLife*. Your revised article has been favorably evaluated by Gary Westbrook (Senior editor), and three reviewers, one of whom is a member of our Board of Reviewing Editors. The manuscript has been improved but there are some remaining very minor issues that need to be addressed before acceptance, as below:

1) To avoid confusion the authors should state in the title of Figure 2 that the results presented are from an "in vitro" preparation

2) The authors maintained the analysis of hypoxia using 10 mm Hg of pO2 threshold for evaluation of nifedipine effect and indicated that this is to focus on the severe type of hypoxia. Although this is acceptable, a minor change in the figure title and corresponding manuscript section title will clarify this point. (e.g., add the word "severe" in the titles).

---

## [Author Response]

*Essential revisions:*

*1) The authors throughout the paper define ischemia as blood flow that leads to less than 10mm of mercury partial pressure of oxygen. This measure results in a threshold change for defining hypoxia. This measurement is potentially misleading because many of the treatments are only partially protective and seizures still produce a reduction in PO2, but not all the way to 10mm Hg partial pressure of O2. To make the manuscript more balanced, the authors should remove this thresholding of hypoxia at 10mm of mercury. In cases where area under the curve is presented, the entire area under the curve should be considered not just whether there is a component, which is below 10mm of mercury. Importantly, the authors were able to perform their studies without the confounding effects of anesthesia, which greatly increases the worth of the study.*

We appreciate the reviewers pointing out that anesthesia could confound both the seizure characteristics as well as the neurovascular response and that our studies in unanaesthetized animals greatly increased its worth.

We understand the concern that defining severe hypoxia as the area below 10 mmHg could be potentially misleading as seizures can still cause a drop in local oxygen levels without crossing that boundary. We have complied with the reviewers’ request to calculate the entire area under the curve between baseline and recovery of pO2 levels for the graphical display of the data and have made those data available with this response. However, we would like to take this opportunity to convince the reviewers that the area below 10 mmHg of pO2 is the more useful and important measure rather than the area below from baseline. We acknowledge that the originally submitted manuscript did not adequately describe the significance of brain pO2 below 10mmHg and we have rectified that situation in our resubmission by expanding the rationale for choosing 10mmHg.

The new section reads, "10mmHg oxygen was chosen as the threshold for defining severe hypoxia since several independent studies have demonstrated that pO2 levels at or below 10mmHg cause significant changes to cellular physiology and brain injury. […]Thus, 10mmHg is a reasonable threshold for defining severe hypoxia and integrating the entire area below 10mmHg combines the depth and duration to reveal the total severe hypoxic burden."

An important strength of the oxygen monitoring methodology used in this manuscript is that it measures the absolute pO2. While relative oxygen recordings can show the magnitude of changes, the importance of absolute pO2 is not properly captured in those measurements. Moreover, the physiological relevance of absolute pO2, particularly below 10mmHg, dramatically outweighs what is learned by measuring relative changes from baseline. For example, if Rat A has a seizure and local brain oxygen decreases from a baseline of 30mmHg to 10mmHg in the postictal period, then Rat A has a net change of 20mmHg. If Rat B has a baseline of 20mmHg and also decreases by 20mmHg, then Rat B is completely anoxic. The total area below baseline pO2 would be identical in Rat A and B (assuming kinetics are the same), but we know from the absolute pO2 values that Rat B is subjected to a much more extreme physiological stressor with molecular and behavioural consequences.

Aside from losing the ability to quantitatively define severe hypoxia, the total area under the curve is particularly susceptible to changes in baseline pO2. We have noted much greater variability in baseline oxygenation (within and between animals) than postictal hypoxia. This variability is seen in our mean oxygen profiles where the error bars at baseline or recovery of oxygenation are 2-3 times greater than during severe hypoxia. This problem introduces a large degree of variability in the area below baseline analysis since baseline oxygen can vary by a wide margin. This becomes especially critical for between subject comparisons where baseline pO2 can vary by more than 15mmHg across animals. For example, the strong positive correlation between seizure duration and area below 10mmHg (Figure 1) is lost when we convert to area below baseline (Figure 7, p=0.99). Instead, area below baseline is substantially dependent on baseline pO2 (p<0.0001). Furthermore, a change small change (3-5mmHg) in the level oxygenation during severe hypoxia has greater physiological significance than at baseline (see additions to manuscript).

Author response image 1.Susceptibility of Area Below Baseline analysis to changes in baseline pO_2_.(**A**) Seizure duration is not correlated with total area under curve (R square=0.000013, p=0.99). (**B**) Instead, the baseline pO_2_ is strongly correlated with the area below baseline (R square=0.52, ***p<0.0001).**DOI:**
http://dx.doi.org/10.7554/eLife.19352.018

We re-analyzed our data to derive the area below baseline to satisfy the reviewers’ interest. We appreciate the reviewers’ concerns to keep this analysis balanced. We believe a threshold of severe hypoxia at 10mmHg is both objective and informative of the true impact postictal changes in oxygenation.

*2) For Figure 2 showing the nifedipine treatment, please calculate panel D based on the entire area under the curve. Furthermore, all statistics in the paper should be based on the total area under the curve and not that below 10mm.*

The reviewers pointed out that a threshold of 10mmHg may be misleading in the case of nifedipine, which is only partially protective. This was not our intention. Pre-treatment with nifedipine works by increasing baseline pO2 as shown in Figure 2. As a thought experiment, if we were to change the threshold to 15mmHg, nifedipine has the ability to inhibit postictal hypoxia (78.82 ± 7.632% reduction of AB10 vs. 69.16 ± 8.577% reduction of AB15 relative to vehicle). However, when we take the entire area below baseline (acknowledging the caveats described above), nifedipine has no effect (Figure 8). This is troubling since administering nifedipine after the seizure dramatically inhibits the area below baseline (Figure 8). Since the area below 10mmHg does not differ between pre- and post-treatment with nifedipine (Table 3), but the area below baseline does, this highlights the susceptibility of area below baseline analysis to changes in baseline pO2.

Author response image 2.Comparing area below baseline analyses with pre- and post-administration of nifedipine.(**A**) Pre-administration of nifedipine had no effect on area below baseline. (**B**) Post-administration of nifedipine significantly reduced the area below baseline (p<0.01).**DOI:**
http://dx.doi.org/10.7554/eLife.19352.019

Considering these drawbacks, we chose to retain our initial analysis (area below 10mmHg) and kept the focus of this manuscript on severe hypoxia following seizures. We have now expanded the discussion to address the issues between relative changes from baseline vs. changes in severe hypoxia (area below 10mmHg) with nifedipine pre-treatment. We hope this acknowledges the reviewers’ concerns and increases the transparency of the data presented.

*3) Simultaneous blood flow and pO2 are shown in Figure 2, this is important as it suggests that the changes in pO2 are secondary to a reduction in blood flow. However, the sample size of 3 for this key measure seems small. We suggest to add more n's and ideally the effect of the various pharmacological manipulations should be extended to blood flow to confirm that these treatments are not just affecting the utilization of O2 by tissue. We would like to emphasize that these experiments should be relatively straight-forward and manageable within a 2 month time window. If the authors cannot perform these experiments, this should be discussed as a limitation in the Discussion section.*

As requested we have added 2 more subjects to Figure 2 which brings the sample size to 5. We have also updated the text to reflect this.

We were unable to perform pharmacological treatments in conjunction with Laser Doppler Flowmetry within the allotted time period. However, it should be noted that we performed experiments with nifedipine (Figure 2) and acetaminophen (Figure 4) in our acute slice preparation to observe their effect on arteriole constriction following seizure stimulation. In both cases, the pharmacological agents prevented vasoconstriction and parallel the in vivo pO2 recordings.

In accordance with the reviewers comments the discussion has now been expanded to address these limitations. It reads, "A limitation of this study is the tool used to measure hypoperfusion in the rat. Laser Doppler flowmetry was used for in vivo recordings, which is an indirect measure of blood perfusion. […] In conjunction with the clinical observations of hypoperfusion, these data support the central role of hypoperfusion in the generation of severe hypoxia in the postictal period."

*4) The data in Figure 2 showing the effects of repeated stimulation on vessel diameter in slices are interesting. However, these effects could potentially be caused by a mechanistically different effect, because the slices have very different conditions than the* in vivo *conditions. Note, that in this case the slices are well-stimulated, but not necessarily ischemic. Moreover, the authors are also unable to determine the identity of the vessels within the slice they are examining. Are these arterioles or veins? Thus, the reviewers are not convinced that the authors can conclude a vasoconstriction mechanism without also adding an* in vivo *measure (2-photon). The authors should discuss this issue as a caveat of the* in vitro *experiments.*

We agree that a limitation of the work in slice is the different conditions that the neurovascular unit is subjected to could complicate the interpretation of the results. We have now addressed this concern in the discussion (see our response to point 3 above). A few things to note are that (1) the degree of vasoconstriction predicts a similar change in perfusion to that seen in vivo and (2) vasoconstriction in slice was sensitive to the same pharmacological agents that prevent hypoxia in vivo (nifedipine and acetaminophen). This provides support that the mechanisms are likely similar in nature.

The reviewers mention that the slices are well-stimulated, but not necessarily ischemic. One of the weaknesses of the manuscript prior to review was that the various methods were not well tied together. In the Results section, we now compare the observations made in slice to those made in vivo. We point out that according to Poiseuille's law a 13.8% vessel constriction translates to a 55.2% reduction in blood flow, which is substantial hypoperfusion. We observed remarkably similar changes in blood flow with laser Doppler flowmetry and confirmed the presence of severe hypoxia under these conditions.

The reviewers raised concerns about the nature of the vessels we were imaging (arterioles or veins?). We only measured vessel diameter in hippocampal arterioles and these vessels were identified by their encapsulation by smooth muscle. This identification feature has been added to the manuscript.

*5) The human data in Figure 3 is interesting, although much of the quantification of these data is in a more descriptive form in the tables. We suggest that the authors present these data as mean levels of blood flow in the different groups before and after seizure. The presentation of the human clinical data is done so in a qualitative manner in the tables. This data should be properly quantified in some form of absolute blood flow units that are comparable to the mm of 02 for the animal work. Please also add some information about how the analysis was done. Were investigators blinded to conditions? What exactly are the controls? Potentially, periods without seizures in the same patients and blood flow at this time could be used as a control. Furthermore, given that some of the patients have multiple seizures, it would also be interesting to know whether the results were repeatable on different imaging sessions.*

We agree and now provide more quantitative analysis of the human blood flow data. We have summarized the quantitative CBF data in Figure 3—figure supplement 1. The data are now presented in two ways: (1) absolute measurements of baseline and postictal blood flow and (2) percent change in blood flow as a function of seizure duration. As requested, the inclusion of percent change makes this data more comparable with the work in animal models. Furthermore, we included the number of each patient beside their correspond data point on each graph to easily allow comparison to the qualitative information presented in the tables.

We have expanded the methods section to include more information on how the analysis was done. Briefly, to obtain the quantitative CBF data, a region of interest (ROI) was manually drawn in a blinded fashion around the single brain area exhibiting the maximal post-ictal blood flow reduction as well as adjacent, confluent voxels with CBF reductions > 10 mL/100g/min. In Table 2, we now provide where the seizure occurred (as determined by an Epileptologist's review of EEG), where maximal hypoperfusion occurred, and the size of the ROI analyzed. We also noted concordance between areas maximally hypoperfused and areas involved in the seizure.

Persons reviewing the ASL data were blinded to the clinical data when determining where the maximal post-ictal CBF changes were seen and when manually drawing ROIs to obtain the quantitative CBF data. Control (baseline) interictal ASL data were obtained from the same patients from whom post-ictal ASL data were obtained. The baseline data were obtained following a seizure-free period of >24 hrs, while the patients were still on the seizure monitoring unit (to confirm that they were in fact seizure-free for > 24 hrs).

We agree with the reviewers that results from multiple seizures would be very useful data to obtain. However, patients are typically discharged from our seizure monitoring soon after obtaining the ASL data. Thus, they are not available for a repeat study while in hospital. It is not possible to obtain postictal ASL data on patients who are outpatients. It is also not possible to re-admit patients to our clinical Seizure Monitoring Unit for research purposes only.

*6) "Significant (> 25 CBF units) postictal hypoperfusion was seen in 7 of 10 subjects (Table 2). Two examples are shown in Figure 3. The location of postictal hypoperfusion was concordant with the clinically determined seizure onset zone in four cases and it was in the same lobe of the brain as the seizure onset in two cases. " Please clarify what is meant by significant hypoperfusion > 25 CBF units? Does this mean a change in perfusion of 25 units? Please explain why >25 CBF units would denote hypoperfusion if we don't know what the upper end of the range could be? The correlation with the clinical data does not seem very strong (2 or 4 of 10 examples?). How does this compare to the O2 values from the animal work? Please add this into the Discussion section.*

Normal cortical cerebral blood flow (CBF) is approximately 60 ml/100g/min. The threshold for which neuronal function is impaired and the tissue is at risk of infarction is approximately 25 ml/100g/min. Given this, we identified the number of patients who had maximal post-ictal CBF reductions of at least 25mL/100g/min compared to baseline since this is likely clinically significant and measurable by ASL. To this end, we identified 7 patients that had maximal postictal CBF reductions of > 25 ml/100g/min. Note, that in this context, the maximal hypoperfusion (> 25 ml/100g/min) described herein would be expected to be higher than the average CBF changes seen in the ROIs that also included adjacent voxels with CBF changes > 10 ml/100g/min (Figure 3—figure supplement 1).

We apologize for the lack of clarity in reporting the concordance of the postictal ASL data with the presumed seizure onset zone. Since the original submission, we have modified our method of determining concordance between the ASL and clinical data and now described it in the methods section.

Unfortunately a direct comparison of blood flow data from humans with pO2 values from the animal models is not possible. We did, however, convert ASL data to percentage decrease in blood flow to be more comparable to rat LDF recordings. The best comparison would be to actually measure blood flow with ASL fMRI in an animal model, whilst simultaneously performing pO2 measures, and then to compare those blood flow measures to those in human epilepsy patients. In fact this is one of our planned experiments for the near future.

[Editors' note: further revisions were requested prior to acceptance, as described below.]

*The manuscript has been improved but there are some remaining very minor issues that need to be addressed before acceptance, as below:*

*1) To avoid confusion the authors should state in the title of Figure 2 that the results presented are from an "*in vitro*" preparation*

We have added "in an in vitro preparation" to the title of Figure 2.

*2) The authors maintained the analysis of hypoxia using 10 mm Hg of pO2 threshold for evaluation of nifedipine effect and indicated that this is to focus on the severe type of hypoxia. Although this is acceptable, a minor change in the figure title and corresponding manuscript section title will clarify this point. (e.g., add the word "severe" in the titles).*

We have made these additions (adding "severe") to both the figure title and corresponding manuscript section titles.